# Climate impact of volcanic eruptions: the sensitivity to eruption season and latitude in MPI-ESM ensemble experiments

Zhihong Zhuo[1, a], Ingo Kirchner[1], Stephan Pfahl[1], Ulrich Cubasch[1]

[1]Institute of Meteorology, Freie Universität Berlin, Berlin, 12165, Germany
[a]now at: Section for Meteorology and Oceanography, Department of Geosciences, University of Oslo, Oslo, 0315, Norway

*Correspondence to*: Zhihong Zhuo (zhihong.zhuo@met.fu-berlin.de)

**Abstract.** Explosive volcanic eruptions influence near-surface temperature and precipitation especially in the monsoon regions, but the impact varies with different eruption seasons and latitudes. To study this variability, two groups of ensemble simulations are performed with volcanic eruptions in June and December at 0° representing an equatorial eruption (EQ) and at 30° N and 30° S representing northern and southern hemisphere eruptions (NH and SH). Results show significant cooling especially in areas with enhanced volcanic aerosol content. Compared to the EQ eruption, stronger cooling emerges in the northern hemisphere after the NH eruption and in the southern hemisphere after the SH eruption. Stronger precipitation variations occur in the tropics than in the high latitudes. Summer and winter eruptions lead to similar hydrological impacts. The NH and the SH eruptions have reversed climate impacts, especially in the regions of the South Asian summer monsoon (SASM). After the NH eruption, direct radiative effects of volcanic aerosols induce changes in the interhemispheric and land-sea thermal contrasts, which move the intertropical convergence zone (ITCZ) southward and weaken the SASM. This reduces the moisture transport from the ocean, and reduces cloud formation and precipitation in India. The subsequent radiative feedbacks due to regional cloud cover lead to warming in India. After the SH eruption, vice versa, a northward movement of the ITCZ and strengthening of the SASM, along with enhanced cloud formation, lead to enhanced precipitation and cooling in India. This emphasizes the sensitivity of regional climate impacts of volcanic eruptions to eruption latitude, which relates to the dynamical response of the climate system to radiative effects of volcanic aerosols and the subsequent regional physical feedbacks. Our results indicate the importance of considering dynamical and physical feedbacks to understand the mechanism behind regional climate responses to volcanic eruptions, and may also shed light on the climate impact and potential mechanisms of stratospheric aerosol engineering.

## 1 Introduction

Sulfate aerosols in the stratosphere from explosive volcanic eruptions significantly cool the surface by reflecting incoming solar radiation (Robock, 2000; Timmreck, 2012). This further affects the earth's hydrological cycle. Both observations and model simulations indicate a reduction of global precipitation after volcanic eruptions (Gillett et al., 2004; Iles and Hegerl, 2014; Iles et al., 2013; Paik and Min, 2016; Robock and Liu, 1994; Trenberth and Dai, 2007). Precipitation was found to

largely decrease in tropical areas (Colose et al., 2016; Iles and Hegerl, 2014), and also notably in summer monsoon regions (Liu et al., 2016; Man et al., 2014; Oman et al., 2006; Schneider et al., 2009; Zambri and Robock, 2016; Zhuo et al., 2020). However, the spatial distribution of volcanic aerosols and associated radiative forcing affects the climate impact of volcanic eruptions (Robock, 2000; Timmreck, 2012; Toohey et al., 2019; Yang et al., 2019). The precipitation response varies with changing distributions of volcanic aerosols between the hemispheres. Inverse effects of interhemispherically asymmetric

volcanic aerosols were found in Sahelian precipitation (Haywood et al., 2013; Jacobson et al., 2020), monsoon climate (Iles and Hegerl, 2014; Liu et al., 2016; Zhuo et al., 2014) and tropical hydroclimate in general (Colose et al., 2016; Zuo et al., 2018). Furthermore, the climate impact of volcanic eruptions is affected by eruption latitude (Marshall et al., 2020; Yang et al., 2019; Zuo et al., 2019) and eruption season (Aquila et al., 2012; Stevenson et al., 2016; Toohey et al. 2011; 2013). Based on volcanic forcing reconstruction indices, most of these previous studies separated historical eruptions into volcanic

classifications with different numbers of events, eruption seasons and unequally distributed aerosol magnitudes in different hemispheres. In order to study the sensitivity of the climate impact of volcanic eruptions to season and location (tropical vs. Northern or Southern Hemisphere) in a systematic way, in this study we thus perform model experiments with artificial volcanic eruptions.

    Tropical eruptions are considered to have larger climate impacts than extratropical eruptions (Myhre et al., 2013; Schneider

et al., 2009). The volcanic aerosols injected into the stratosphere from a tropical eruption can be transported to both hemispheres and finally reach both poles (Robock, 2000; Aquila et al., 2012). This generates larger areas being affected by aerosols with long lifetimes in the stratosphere, thus causing larger and longer-lasting climate effects (Schneider et al., 2009). Based on this hypothesis, previous reconstructions of volcanic forcing assumed a longer lifetime of aerosols injected by tropical volcanic eruptions than by extratropical eruptions (Ammann and Naveau, 2003; Gao et al., 2008). Thus, compared to

tropical eruptions, model results suggested shorter cooling periods and less severe reduction of tropical precipitation after high-latitude eruptions (Schneider et al., 2009). On the contrary, using both ice-core and tree ring-based proxy reconstruction and model simulations, Toohey et al. (2019) found larger climate impact after extratropical eruptions than tropical eruptions. They pointed out that the overestimated volcanic forcing from tropical eruptions results in an overestimation of their climate impacts. Thus, previous studies came to different conclusions on whether tropical or extratropical volcanic eruptions have

larger climate impact.

    Only few studies investigated the mechanism behind the precipitation response to volcanic eruptions (Man and Zhou, 2014; Man et al., 2012, 2014; Oman et al., 2006; Paik and Min, 2016). Paik and Min (2016) argued that a reduced vertical motion after volcanic eruption induces the reduction of global precipitation. The reduction of the summer monsoon precipitation was suggested to be due to a decreased land-sea thermal contrast and the subsequent weakening of the summer monsoon (Man

and Zhou, 2014; Man et al., 2012, 2014; Oman et al., 2006). Conclusions in these studies did not consider regional differences and feedback processes in the Indian monsoon region (Oman et al., 2006; Paik and Min, 2016). The inversed climate impact after interhemispherically asymmetric volcanic aerosol injections was suggested to result from a displacement of the ITCZ towards the warmer hemisphere with less volcanic aerosol loading (Colose et al., 2016; Haywood

et al., 2013; Iles and Hegerl, 2014; Zuo et al., 2018). Here, we study the mechanisms behind the regional climate response to volcanic eruptions at different latitudes and in different seasons, focusing on the Indian monsoon region, with the help of a series of model simulations. This study aims to answer the following questions: how are global and regional climate impacts of volcanic eruptions affected by the eruption season and latitude? Do tropical eruptions have larger climate impact than extratropical eruptions? What is the mechanism behind the Indian monsoon response to volcanic eruptions at different latitudes?

After this introduction, we describe the methods including model description and experimental setup in section 2. In section 3, we present our results and discussion. We first show the global forcing and climate responses to volcanic eruptions in section 3.1 to 3.4, and then focus on the mechanism of the precipitation response in India in section 3.5. Summary and conclusions are given in section 4.

## 2 Methods

### 2.1 Model description

We perform simulations with the Max-Planck-Institute Earth System Model (MPI-ESM) (Giorgetta et al., 2013). The MPI-ESM is a fully coupled general circulation model with ECHAM6 as atmosphere component (Stevens et al., 2013), MPIOM as ocean component (Jungclaus et al., 2013), JSBACH for simulating the terrestrial biosphere (Reick et al., 2013; Schneck et al., 2013) and HAMOCC5 for the biogeochemistry of the ocean (Ilyina et al., 2013). The components are coupled through the exchange of energy, momentum, water and carbon dioxide using the OASIS3 coupler (Valcke, 2013). In this study, we use the low-resolution (LR) configuration of the model. In the atmospheric component, it has a horizontal resolution of T63 with 47 vertical levels extending to 0.01 hPa, thus including the stratosphere. The oceanic component has a resolution of 1.5° (near the equator) with 40 vertical levels.

Also because of its high simulation efficiency (Roeckner et al., 2006), the older MPI-ESM version with ECHAM5 as its atmospheric component has been extensively used to study the Asian summer monsoon (Guo et al., 2016; Man et al., 2012) and the climate response to volcanic eruptions (Man et al., 2014; Zhang et al., 2012). Studies on the impact of volcanic eruptions have also been performed with the newest model version that is also applied here (Timmreck et al., 2016; Toohey et al., 2016).

We use the same configuration of the MPI-ESM model as used for the historical simulation of CMIP6. Chemical processes are not explicitly simulated, and background tropospheric and stratospheric aerosols, as part of the forcing data, are represented by their aerosol optical properties (Giorgetta et al., 2013). The volcanic forcing used in this study is produced by the easy volcanic aerosol (EVA) module (Toohey et al., 2016). Through setting the eruption year, month, sulfate injection magnitude and hemispheric ratio in EVA, stratospheric sulfate mass injected by the volcanic eruption is transferred to mid-visible (550 nm) aerosol optical depth (AOD) and effective radius as volcanic forcing input for model simulations. Pinatubo-like eruptions are simulated because the construction of EVA relies extensively on observational constraints, especially the

observational records of the 1991 Pinatubo eruption. Other external forcings for MPI-ESM are the same as in the CMIP6 historical experiment, except that the anthropogenic emissions are fixed at the 1900 level.

## 2.2 Experimental setup

In order to study the climate impacts of volcanic eruptions at different eruption latitudes and in different seasons, two groups

of three different latitudinal volcanic eruptions in summer and winter are simulated (here summer and winter refer to the boreal seasons). Following Toohey et al. (2016), 9 Tg of total sulfur injection magnitude is prescribed. The eruption latitudes are set to be 0° for the equatorial eruption (EQ case) and 30° N and 30° S for the northern and southern hemispheric eruptions (NH and SH cases), respectively. For the summer eruptions, the date is set to be the same as the 1991 Pinatubo eruption on June 15, 1991; for the winter eruptions, the date is set to be December 15, 1991. We perform 10-member

simulations for each eruption case.

Figure 1 shows the experimental design of our simulations. An initial run is performed for the period of 1966-2012, and then we perform 23 control runs for the period of 1986-1995 without any volcanic eruption. For these 23 members, we calculate the Oceanic Niño Index (ONI) to quantify El Niño-Southern Oscillation (ENSO) variability. ONI values between -1 and 1 represent neutral or weak ENSO conditions. The 1991 Pinatubo eruption was accompanied by a strong El Niño event in

1991-1994. The response of ENSO to volcanic eruptions has been widely investigated (Adams et al., 2003; Khodri et al., 2017; Stevenson et al., 2016) but remains an unsolved question. To rule out the concurrent effect of ENSO, we pick out 10 control runs with neutral or weak ENSO in the period of 1990-1992 as a basis for our experiments. Restart files from these 10 control runs are used to initialize six 10-member ensemble simulations of EQ, NH and SH eruptions in summer and winter for the period of 1991-1996 and 1990-1996, respectively. Note that the simulated years do not correspond to the real

115   years because of the free-running spin-up simulations.

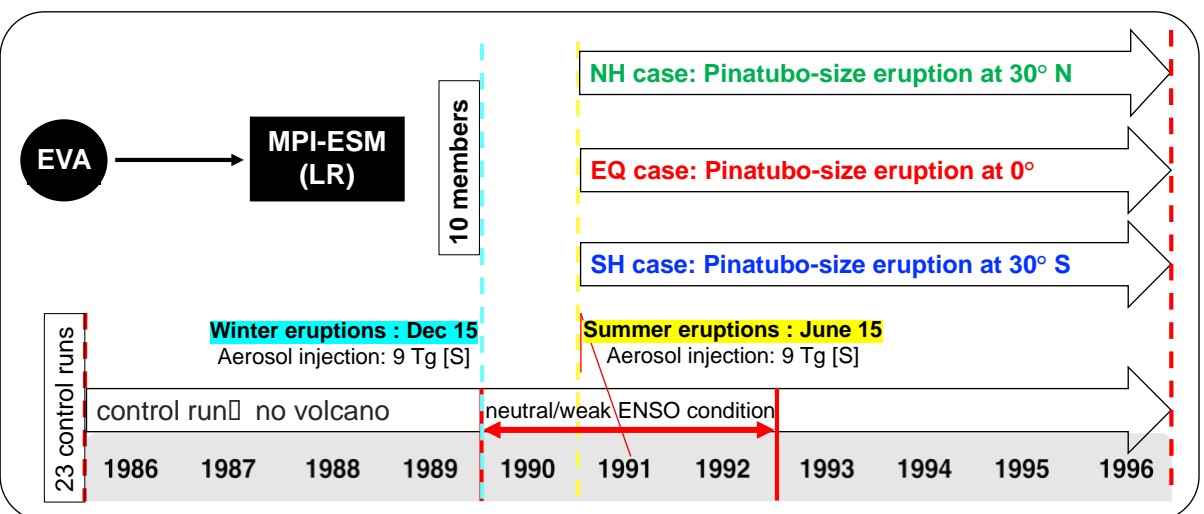

**Figure 1.** Schematic view of the experimental design. Simulations are performed using MPI-ESM (Giorgetta et al., 2013) with volcanic forcing produced from the EVA module (Toohey et al., 2016).

## 2.3 Analysis methods

The position of the ITCZ is indicated by the latitude of the maximum zonal mean precipitation between 20ºN and 20ºS. To quantify the Indian summer monsoon response, we calculate the south Asian summer monsoon index (SASMI), which is defined as the difference between the zonal wind at 850 hPa over the region 0º-20ºN, 40ºE-110ºE and the zonal wind at 200 hPa over the region 0º-20ºN, 40ºE-110ºE (Webster and Yang, 1992).

To study horizontal moisture transport, we calculate the vertically integrated moisture transport (IVT) and its divergence (IVTD). The IVT is calculated using the following equation:

$$(1/g) \int_{surface}^{model\ top} q\vec{v}\ dp$$

where g is the acceleration due to gravity, $q$ is specific humidity, $\vec{v}$ is the horizontal wind vector, and $p$ is pressure. The vertical integration of the equation is performed from the surface to the model top.

We calculate the multi-member mean (MEM) of adopted variables and the indices defined above to study the hydrological effect of the summer and winter volcanic eruptions at different latitudes. Both temporal and spatial analyses are performed to investigate the climate effects. Anomalies with respect to the MEM of the control runs without any volcanic eruption are presented. The standard deviation (SD) of the control runs is calculated to indicate the significance of the temporal analysis results. Two-tailed student-t tests at the 95% and 99% confidence levels are performed for testing the significance of the spatial results.

## 3 Results and discussion

### 3.1 Volcanic forcing from EVA

The generated volcanic forcing is indicated by the aerosol optical depth at 550 nm ($AOD_{550}$). Figure 2a and 2b show the temporal variation of global mean $AOD_{550}$ in 1991-1996 and 1990-1996 for the summer and winter eruptions, respectively. All three eruption cases in each season lead to identical time series of global mean $AOD_{550}$, with a steep increase in the beginning due to the formation of sulfate aerosols after the volcanic eruptions. The highest global mean $AOD_{550}$ of 0.12 is identical in all the cases, and is reached after six months, i.e. in December 1991 and June 1991 for the summer and winter eruptions, respectively. Figure 2c and Fig. 2d show the spatio-temporal structure of the volcanic forcing. The formation and distribution of volcanic aerosols take several months. For the EQ eruption cases, the $AOD_{550}$ indicates that volcanic aerosols are transported to both hemispheres, associated with a stronger dispersion than in the NH and SH cases, and more aerosols are transported to the northern hemisphere in the winter case and to the southern hemisphere in the summer case. This indicates that the transport of volcanic aerosols from equatorial eruptions to high latitudes depends on the eruption season, which is related to the large-scale transport of the Brewer Dobson circulation (Hamill et al., 1997). For the NH and SH

eruption cases, volcanic aerosols are mostly confined to the specific hemisphere, with large AOD$_{550}$ in the subtropical areas between 30° and 60° latitude.

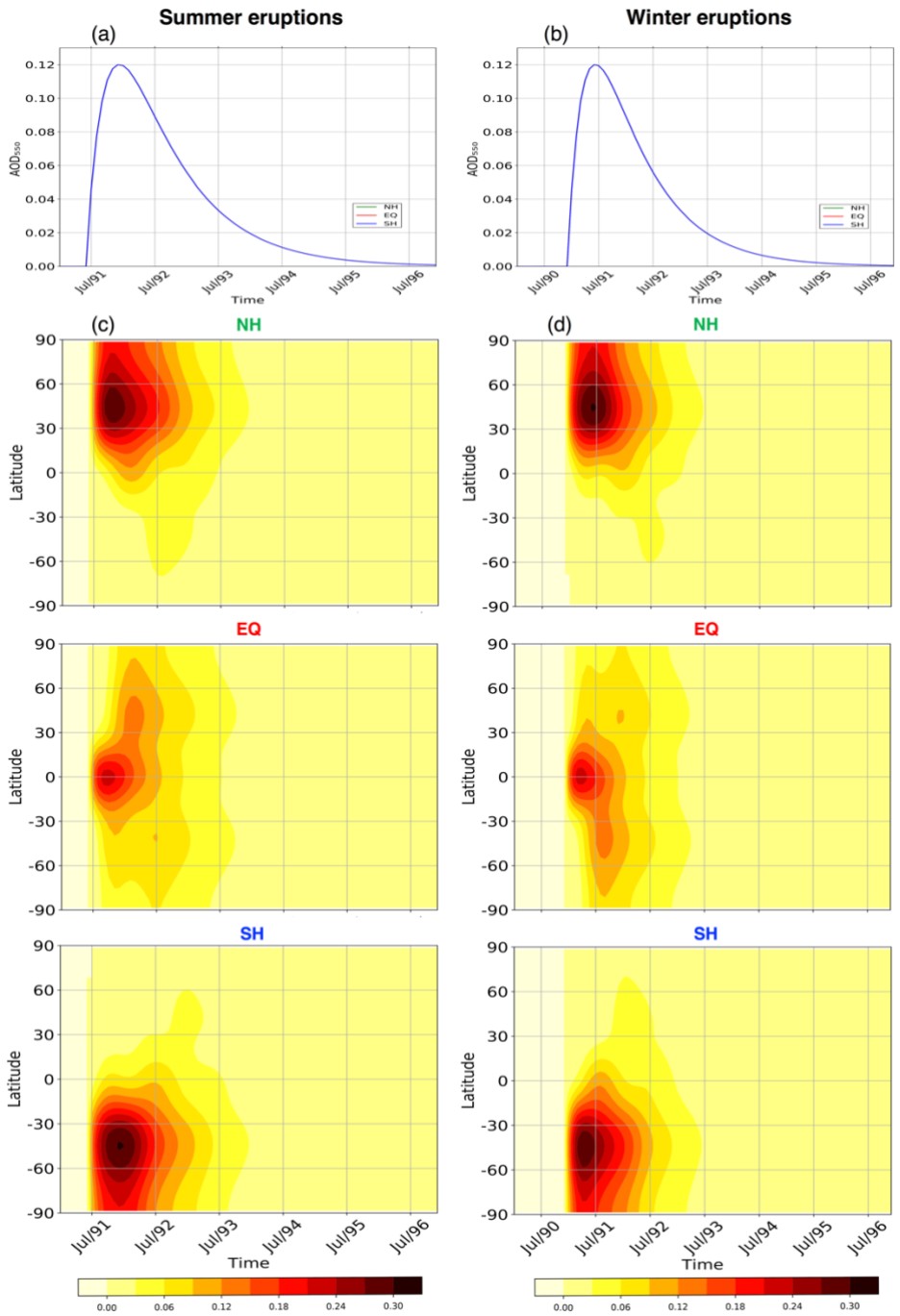

**Figure 2.** Global mean aerosol optical depth anomaly at 550 nm for the summer (a) and winter (b) volcanic eruption cases and their zonal mean distributions (c and d).

### 3.2 Solar radiation response

To show the primary radiative effects of the volcanic aerosols, we present the net outgoing shortwave radiation (OSR) at the top of the atmosphere (TOA) in clear-sky conditions. Figure 3 shows the time series of the global mean OSR anomaly and its zonal mean distribution. The OSR value in the MPI-ESM is negative, indicating an increased OSR after the volcanic eruptions. For all three cases of summer eruptions (Fig. 3a), the increased OSR lasts for three years until 1994. Although the global mean $AOD_{550}$ is identical among the three cases (Fig. 1a), the OSR increases continuously after the SH and the EQ eruptions, but after the NH eruption, there is a slight decrease of OSR in the boreal winter of 1991. Similar response patterns are found for the winter eruption cases (Fig. 3b), except that the slight decrease of OSR occurs after the SH eruption case in the boreal summer of 1992, which is the austral winter of the southern hemisphere. This relates to the seasonal change of the incoming solar radiation in the different hemispheres. The zonal mean distribution of the OSR anomaly in all summer cases (Fig. 3c) and winter cases (Fig. 3d) indicates a simultaneous increase of OSR in the areas with high AOD after the eruption. This indicates that more solar radiation is reflected in regions with more volcanic aerosols. For the summer eruption cases (Fig. 3c), the shape of the OSR zonal mean distribution is similar to that of $AOD_{550}$ in the EQ and SH cases, while OSR is reduced in the NH case in the boreal winter of 1991, resulting from the reduced incoming radiation in winter. For the winter eruption cases (Fig. 3d), similar shapes between OSR and $AOD_{550}$ are shown in the EQ and NH cases, while a reduced OSR in the austral winter of 1991 is found in the SH case. This reflects the role of the seasonal change of the incoming solar radiation in the two hemispheres.

### 3.3 Temperature response

SR is reflected by the volcanic aerosols in the stratosphere. This decreases the SR reaching the surface and results in surface cooling. Figure 4 shows the time series of global mean surface temperature (T) anomaly and its zonal mean distribution. Significant cooling occurs after the volcanic eruptions. For the summer eruption cases, it takes 15, 12 and 15 months to reach the maximum cooling after the NH, the EQ and the SH eruption, respectively, thus stronger cooling emerges after the NH eruption than the SH and EQ eruptions, and the coolest boreal summer among all the cases occurs in 1992 (Fig. 4a). Most significant cooling emerges in the northern hemisphere mid-latitudes in the NH case, in the tropics in the EQ case and in the southern hemisphere mid-latitude areas in the SH case, as indicated by the stippling in Fig. 4c. For the winter eruption cases, a significant and strong cooling is visible in the boreal summer of 1991; the maximum cooling shows 11 months after the NH eruption (Fig. 4b) and emerges in the northern hemisphere mid-latitude areas (Fig. 4d). The maximum cooling occurs 16 months after the EQ eruption in the boreal spring of 1992 (Fig. 4b), and the abnormal cooling is pronounced in the tropics as indicated by the stippling in Fig. 4d. A similar cooling trend with a smaller magnitude is shown 16 months after the SH eruption (Fig. 4b), but the cooling is significant in the southern hemisphere mid-latitudes (Fig. 4d). For both the summer and winter eruptions, the largest cooling occurs several months later than the solar radiation variation but in the corresponding areas with more reflected SR by volcanic aerosols, and the response is faster in the NH cases than in the SH and EQ cases.

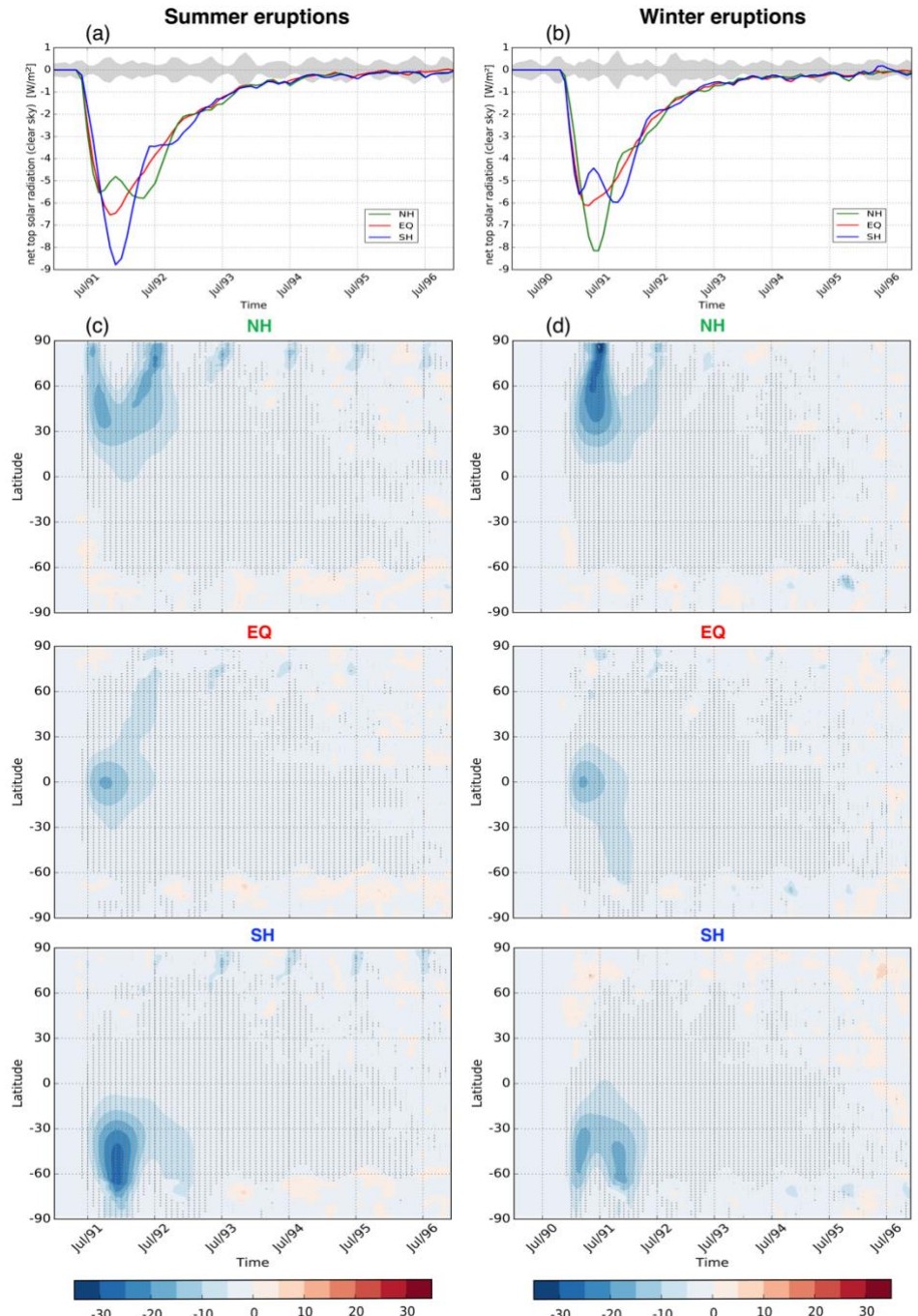

**Figure 3.** Global mean net outgoing solar radiation (W/m²) anomaly at top of the atmosphere in clear-sky conditions for the summer (a) and winter (b) volcanic eruption cases and their zonal mean distributions (c and d). The grey shading in (a) and (b) indicates two standard deviation of the control runs without any volcanic eruption. Dark grey and light grey stippling in (c) and (d) indicate the grid points with significant differences based on the two-tailed student's t-test at the 95% and 99% confidence level, respectively.

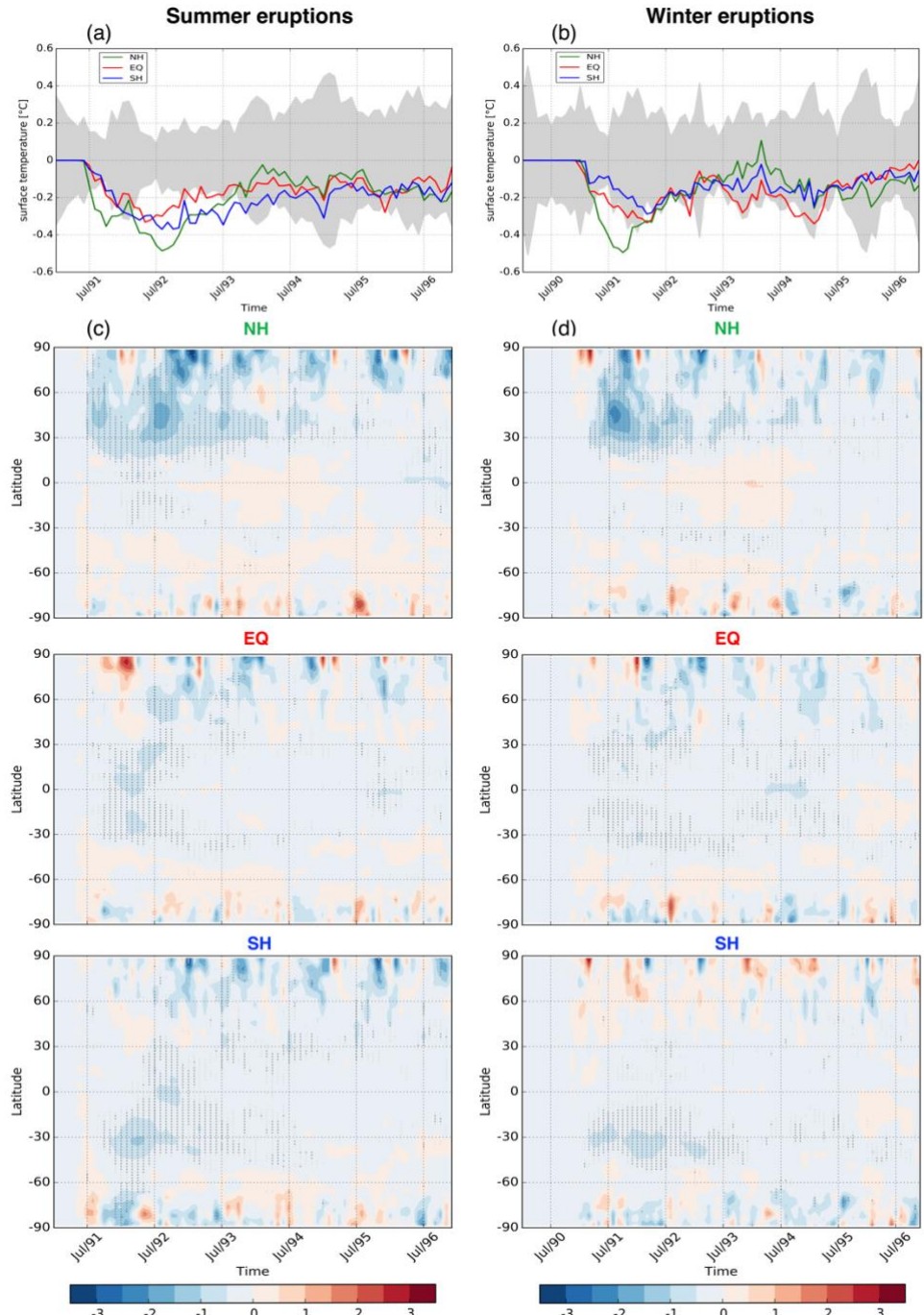

**Figure 4.** Global mean surface temperature (ºC) anomaly for the summer (a) and winter (b) volcanic eruption cases and their zonal mean distributions (c and d). The grey shading in (a) and (b) indicates two standard deviations around the mean of the control runs without any volcanic eruption. Dark grey and light grey stippling in (c) and (d) indicate the grid points with significant differences based on the two-tailed student's t-test at the 95% and 99% confidence level, respectively.

These different responses originate from the delayed response in the ocean compared to the land, because the ocean has a larger heat capacity and content compared to the land and the NH has a larger landmass compared to the SH.

We further analyze the spatial distribution of the temperature response. The time series and zonal mean distribution have shown that the strongest cooling occurs in summer 1992 for the summer eruptions and in summer 1991 for the winter NH eruption and spring 1992 for the winter EQ and SH eruptions. Since we want to discuss more about the summer climate response especially in the northern hemisphere, for the spatial distribution of temperature, we focus on the JJA mean of 1992 for the summer eruptions and 1991 for the winter eruptions (Fig. 5). For the NH summer eruption case (Fig. 5a), significant cooling (99% confidence level) is found for most of the land areas in the northern hemisphere, except for India and the Sahel, where positive temperature anomalies prevail. Stronger cooling occurs in the northern hemisphere than in the southern hemisphere. Over the ocean, most of the areas experience cooling in the northern hemisphere but warming in the southern hemisphere. While similar patterns with stronger magnitude are found especially in the northern hemisphere for the NH winter eruption case (Fig. 5b), different responses are shown over the southern hemispheric ocean and Antarctica. For the SH summer eruption case (Fig. 5e), significant cooling (99% confidence level) is found in the southern hemispheric land areas, but warming emerges in part of the northern hemispheric land areas. Opposite to the NH eruption case, cooling occurs in Indian and the Sahel. Over the ocean, cooling prevails in the low latitude areas while different warming and cooling emerges in different areas in mid and high latitudes. Similar response patterns are found in most of the southern hemispheric land areas in the SH winter eruption case, but warming prevails in most of the northern hemispheric land areas except for the cooling in India and the Sahel (Fig. 5f). For the EQ cases (Fig. 5c and 5d), cooling occurs in most of the land areas, but weaker in magnitude in the northern and southern hemisphere compared to the NH and SH eruption cases, respectively. This indicates that, compared to tropical eruptions, extratropical volcanic eruptions have stronger cooling effects in the hemisphere where the eruption occurs due to concentration of volcanic aerosols. Furthermore, the responses in the low latitudinal land areas in the EQ summer and winter eruption case (Fig. 5c and 5d) are similar to those in the SH summer and winter eruption case (Fig. 5e and 5f).

### 3.4 Precipitation response

The radiative effect of volcanic aerosols, in addition to temperature variations, also leads to changes in the hydrological cycle. Time series of global mean precipitation from our experiments do not show significant changes (not shown), but there are precipitation responses on a regional level. Figure 6 shows the spatial distribution of the precipitation response to the volcanic eruption cases. Generally, stronger absolute precipitation responses occur in the tropics than in the extratropics as well as over the ocean compared to over land. For each eruption latitude, the precipitation response patterns are similar between the summer eruptions (Fig. 6 left panel) and the winter eruptions (Fig. 6 right panel). In many tropical regions, the NH (Fig. 6a and 6b) and SH (Fig. 6e and 6f) volcanic eruptions lead to reversed precipitation responses, i.e. opposite decrease and increase of precipitation up to over 3 mm/day in some areas. Precipitation response patterns in the EQ eruption

cases (Fig. 6c and 6d) are close to those in the SH eruption cases (Fig. 6e and 6f). The most pronounced changes over land are found in India, which is one of the most typical monsoon regions. Precipitation is reduced in India after the NH eruptions (Fig. 6a and 6b), but strongly increased after the EQ and SH volcanic eruptions (Fig. 6c to 6f).

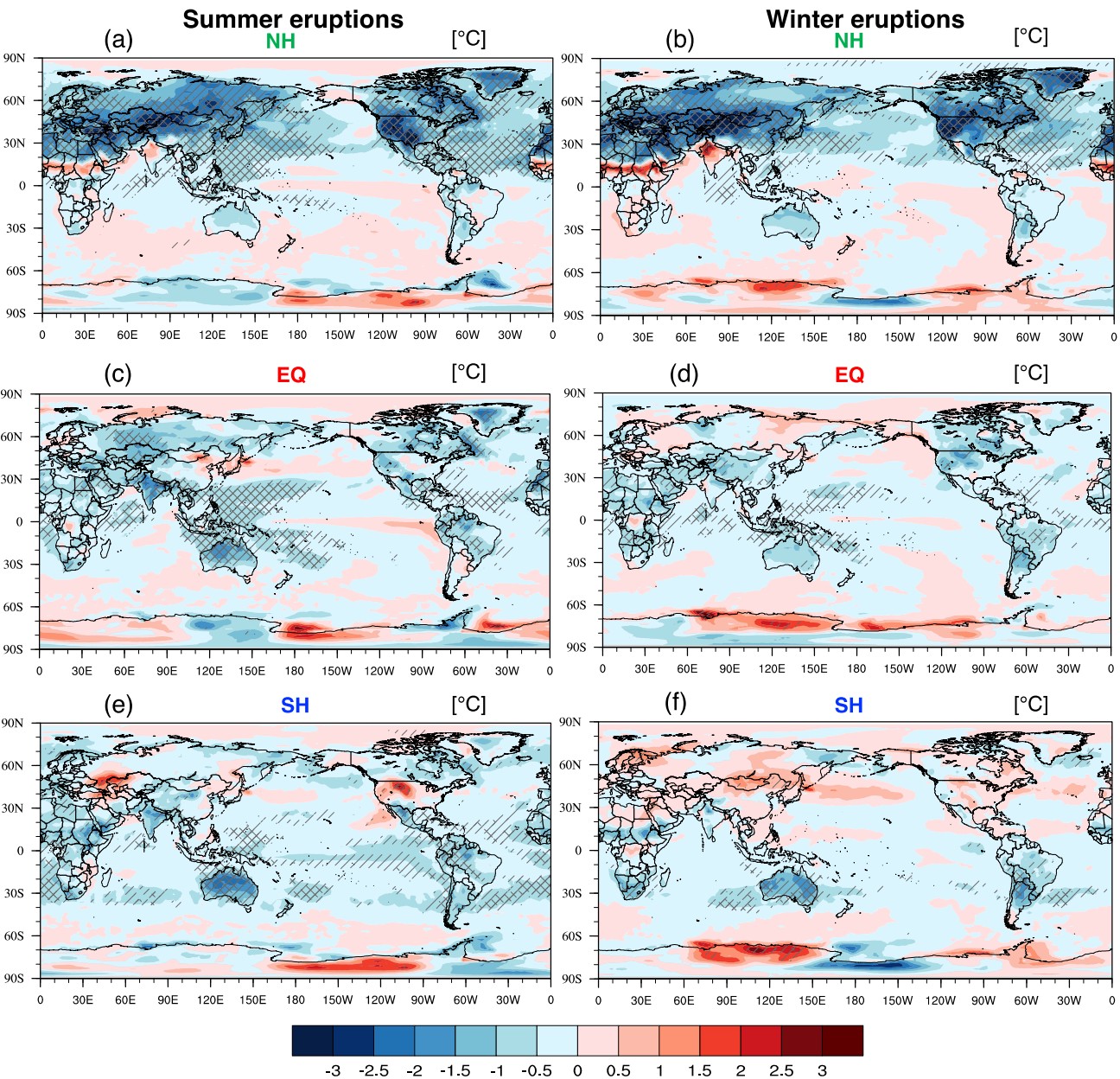

**Figure 5.** Spatial distribution of JJA mean temperature (°C) anomaly in 1992 for the summer NH (a), EQ (c), SH (e) eruption cases and in 1991 for the winter NH (b), EQ (d) and SH (f) eruptions cases. The cross signs and slashes indicate the grid points with significant differences based on the two-tailed student's t-test at the 95% and 99% confidence level, respectively.

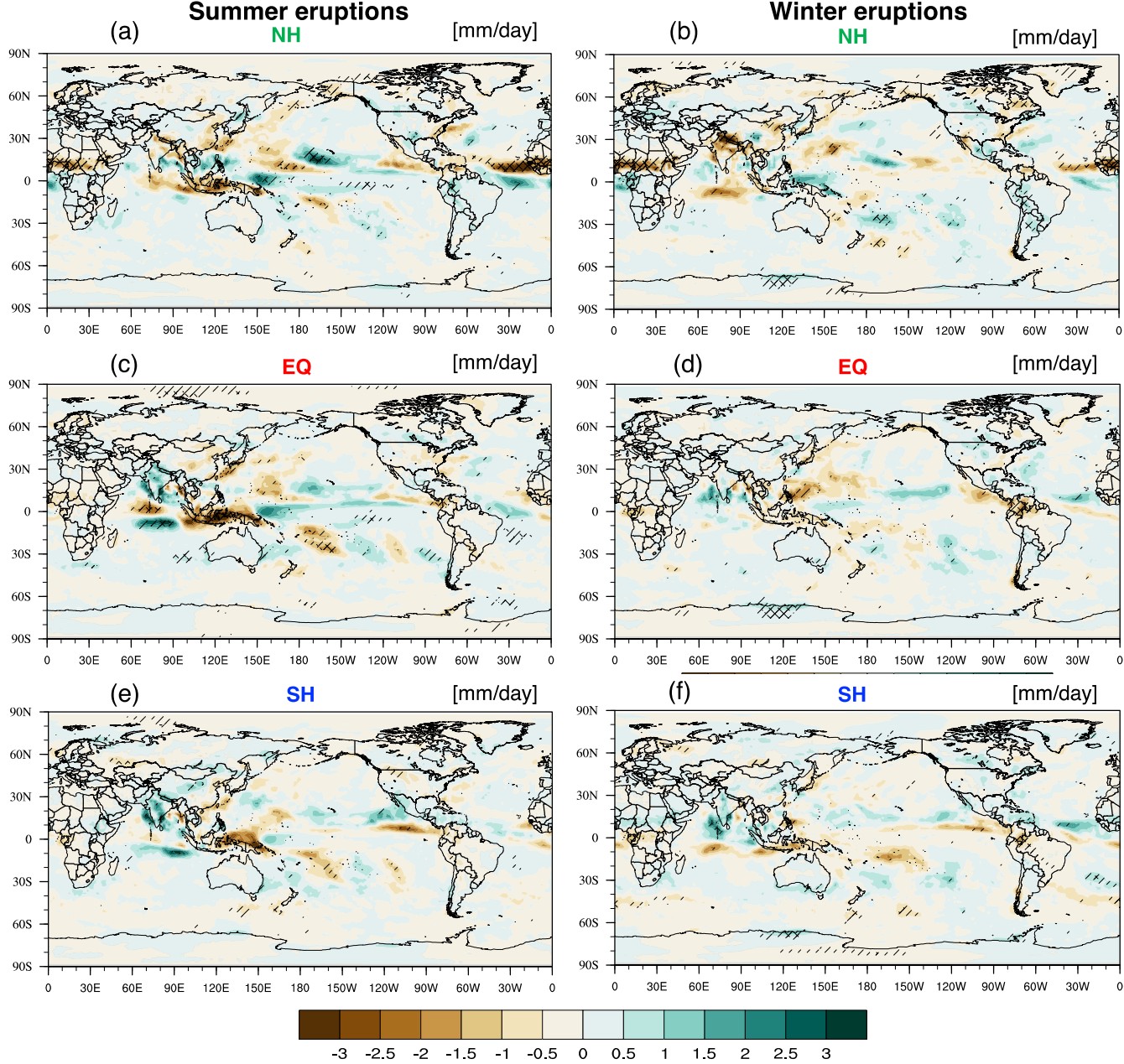

**Figure 6.** Spatial distribution of JJA mean precipitation (mm/day) anomaly in 1992 for the summer NH (a), EQ (c), SH(e) eruption cases and in 1991 for the winter NH (b), EQ (e) and SH (f) eruption cases. The cross signs and slashes indicate the grid points with significant differences based on the two-tailed student's t-tests at the 95% and 99% confidence level, respectively.

### 3.5 Mechanism of precipitation response in India

The temperature and precipitation response patterns clearly indicate a tendency towards inverse climate responses to the NH and SH volcanic eruptions, which are particularly strong in India. The following section thus aims to explain the mechanism behind the different climate effects of asymmetrically distributed volcanic aerosols in India.

### 3.5.1 The direct radiative effect and dynamical response

Sulfate aerosols in the stratosphere from explosive volcanic eruptions reflect solar radiation at the top of the atmosphere (Fig.
3), thus leading to significant surface cooling (Fig. 4). These cooling effects are more pronounced in the areas with more aerosols. Volcanic aerosols are asymmetrically distributed after the different latitudinal volcanic eruptions (Fig. 1). This causes asymmetric cooling effects between the two hemispheres. Figure 7a shows the difference of the boreal summer mean temperature anomaly between the northern and the southern hemisphere in the three summer eruption cases. It decreases largely in 1992 after the NH volcanic eruption and decreases slightly after the EQ eruption, while it keeps close to zero after
the SH volcanic eruption. This indicates a larger cooling in the northern hemisphere than in the southern hemisphere after the NH and the EQ volcanic eruption, while only a slight difference is shown between the two hemispheres after the SH volcanic eruption. Similar results are found in the winter NH and EQ eruption cases, while the temperature difference between the hemispheres increases slightly in the SH case, indicating a larger cooling in the southern hemisphere than in the northern hemisphere after the winter SH volcanic eruption (Fig. 7b).
The thermal contrast between the two hemispheres moves the ITCZ away from the cooler hemisphere (Broccoli et al., 2006). As shown by the yellow line in Fig 7c. and Fig 7d, in boreal summer, large amounts of precipitation are concentrated around 10°N, which is the mean latitude of the ITCZ. After the NH summer volcanic eruption (Fig. 7c), the zonal mean precipitation decreases around 10°N but increases around 0°. This indicates that the ITCZ moves southward toward the equator. The zonal mean precipitation increases north of 10°N but decreases around the equator after the SH summer eruption, which indicates a
northward movement of the ITCZ. For the EQ summer eruption case, because the aerosols are transported to both hemispheres, there is no simple displacement of the ITCZ, but a more complex change with a precipitation decrease around 10°N but increases to the north and south of 10°N. Similarly, a southward shift of the ITCZ is found after the NH winter eruption while a northward shift is found after the SH winter eruption (Fig. 7d). For the EQ winter eruption case (Fig. 7d), the zonal mean precipitation slightly increases around 10°N, indicating a slight northward movement of the ITCZ. The
movement of the ITCZ particularly affects the climate in India.
Due to the large heat content of the ocean, the temperature response over the ocean is attenuated compared to the response over land. This leads to a stronger cooling of the land compared to the ocean, which decreases the land-ocean thermal gradient in summer and leads to a weakening of the summer monsoons (Dogar and Sato, 2019; Iles and Hegerl, 2014; Man and Zhou, 2014; Zuo et al., 2019). Here, we quantify the Indian summer monsoon response to volcanic eruptions with the
265 SASMI (Webster and Yang, 1992). As shown in Fig. 8a, the SASMI decreases continuously until 1994 after the NH summer

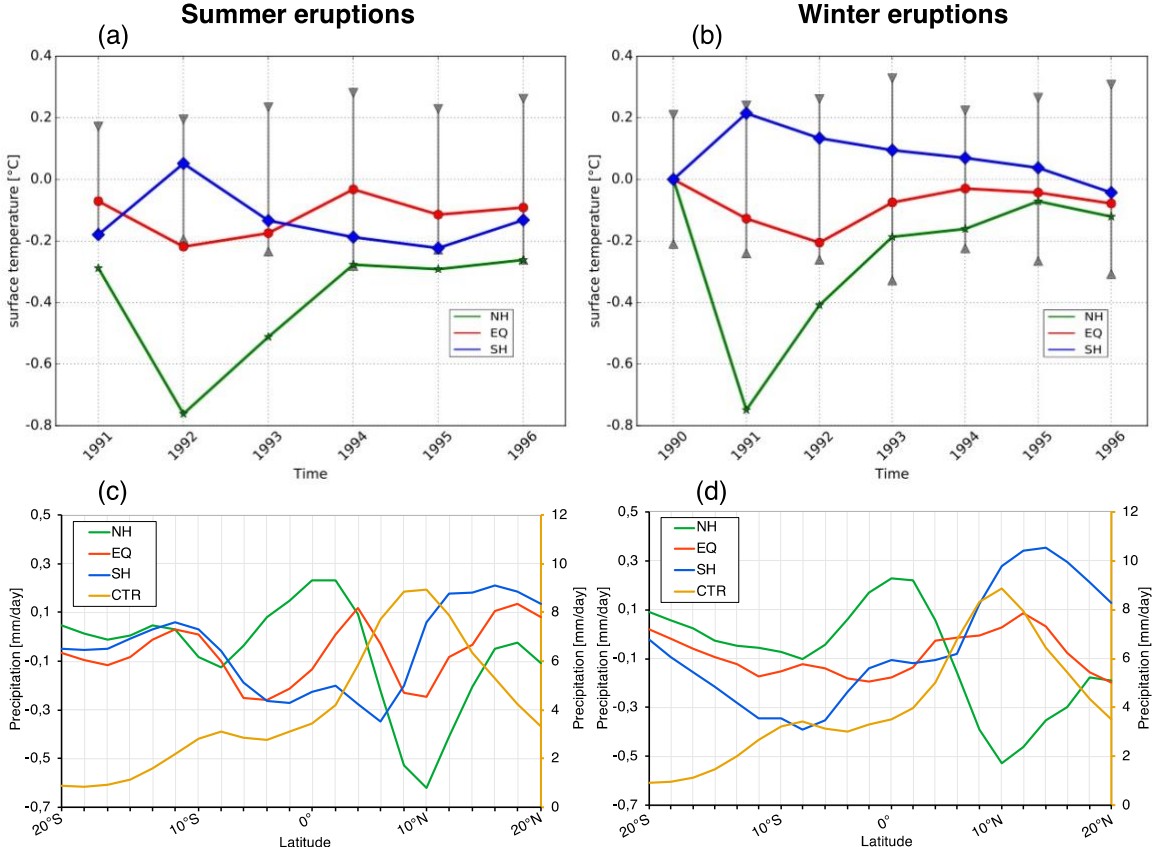

**Figure 7.** Difference of JJA mean temperature (°C) anomaly between the northern and the southern hemisphere after the summer eruptions (a) and the winter eruptions (b), and the position of the ITCZ in 1992 (c) and 1991 (d) indicated by the JJA mean zonal mean precipitation (mm/day) anomaly between 20°N and 20°S. The yellow line (right axis) in (c) and (d) indicates the position of the ITCZ for the control runs, which is calculated with raw precipitation data. The grey bar in (a) and (b) indicates the two-standard deviation of the control runs.

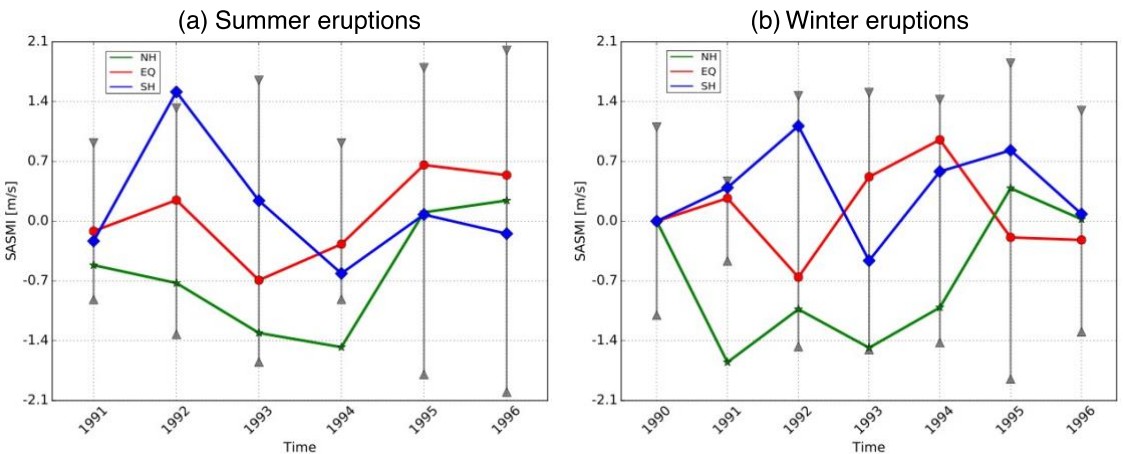

**Figure 8.** South Asian summer monsoon index (m/s) anomaly for the summer eruption cases (a) and the winter eruption cases (b). The grey bar indicates one standard deviation of the control runs.

eruption, while it increases strongly in 1992 after the SH summer eruption and increases slightly after the EQ summer eruption. Similarly, in Fig. 8b, a significant decrease of the SASMI is found in 1991 after the NH winter eruption, while a slight increase of the SASMI occurs in 1991 after the SH and EQ winter eruptions. The SASMI keeps increasing in 1992

after the SH winter eruption. This indicates opposite weakening and strengthening effects of the different NH and SH volcanic eruptions on the south Asian summer monsoon: NH eruptions preferentially cool the NH land regions and tend to weaken the Indian monsoon, while SH cooling following SH eruptions leads to a stronger monsoon circulation.

The altered land-ocean thermal contrast and monsoon circulation largely change the moisture transport from the ocean to India. Because the temperature and precipitation responses, the ITCZ movement and the SASMI all show similar patterns

between the summer eruption cases and the winter eruption cases, we only use the summer eruption cases to explain the response mechanism in India in the following. Figure 9 shows the vertically integrated moisture transport (IVT, vector) and its divergence (IVTD, shaded) over India. In the control runs without any volcanic eruption, as shown in Fig. 9a, in the western part of India, winds from the Arabian Sea bring large amounts of moisture to the land. This results in strong precipitation along the west coast of India. In the eastern and northern parts of India, the precipitation results from the

convergence of southeasterly moisture transported from the Bay of Bengal vortex as shown by the IVTD (green shade). After the NH volcanic eruption, the moisture transported from the Bay of Bengal to the northern part of India is largely reduced, associated with reduced convection and upward motion, as shown by the reduced convergence (brown shade in Fig. 9b). On the contrary, after the EQ and the SH volcanic eruptions (Fig. 9c and 9d), more moisture is transported to the land from both the Arabian Sea and the Bay of Bengal. Moisture convergence is reduced in the northern part of India (brown

shade), while it is strengthened in the southwest and southeast coast of India (green shade). The altered horizontal and vertical motion of the atmospheric circulation thus change the amount and distribution of atmospheric moisture as well as the precipitation patterns over India.

These results show a dynamical response of the climate system to the radiative effect of volcanic aerosols. Changes in the available energy and thermal gradients between the hemispheres move the ITCZ southward after the NH volcanic eruption

but northward after the SH and EQ eruptions (Fig. 7b), which largely affects regional precipitation in the areas along the ITCZ. After the NH volcanic eruption, the altered land-ocean thermal gradients reduce the monsoon circulation and thus the moisture transport and convection over the western coast of India and central India, which, oppositely, is strengthened after the SH and EQ eruptions. These dynamical responses influence the regional climate in India and lead to a precipitation decrease after NH eruptions, but an increase following SH eruptions (see again Fig. 6).

**3.5.2 Radiative feedbacks due to changes in cloud cover**

In the following, physical feedback mechanisms associated with changes in cloud cover and the corresponding radiative effects are discussed, which additionally influence the regional temperature and precipitation variations. Figure 10 shows the net surface solar radiation distribution in clear-sky and all-sky conditions. In clear-sky conditions without taking clouds into consideration (Fig. 10a), the surface solar radiation is generally reduced with a zonally rather symmetric distribution. The

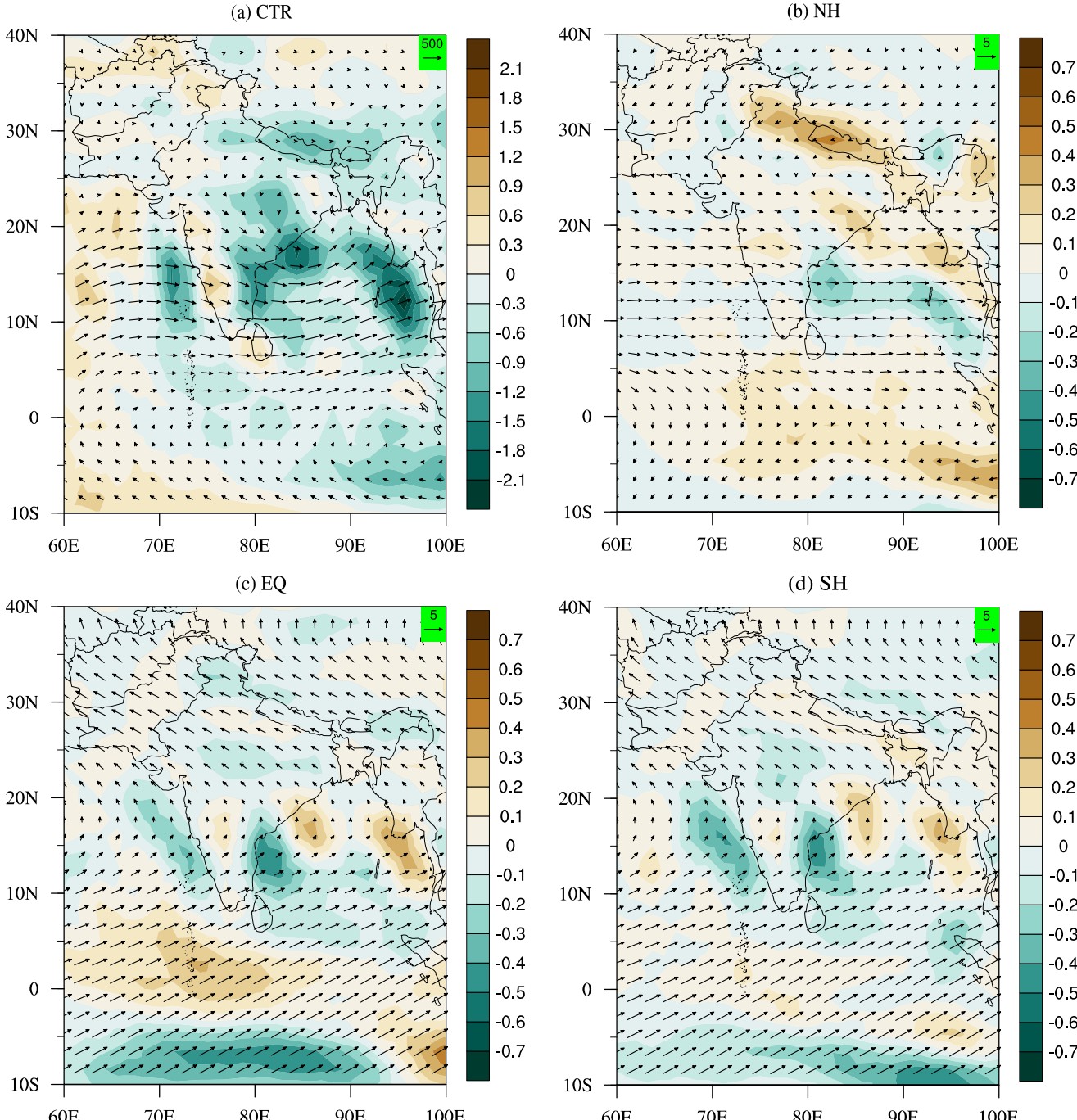

**Figure 9.** JJA mean vertically integrated moisture transport (IVT, vector, kg/m·s) and its divergence (IVTD, shaded, kg/m2·s) for the control runs without any volcanic eruption (a), and their anomalies after the NH (b), EQ (c) and SH (d) volcanic eruptions in 1992.

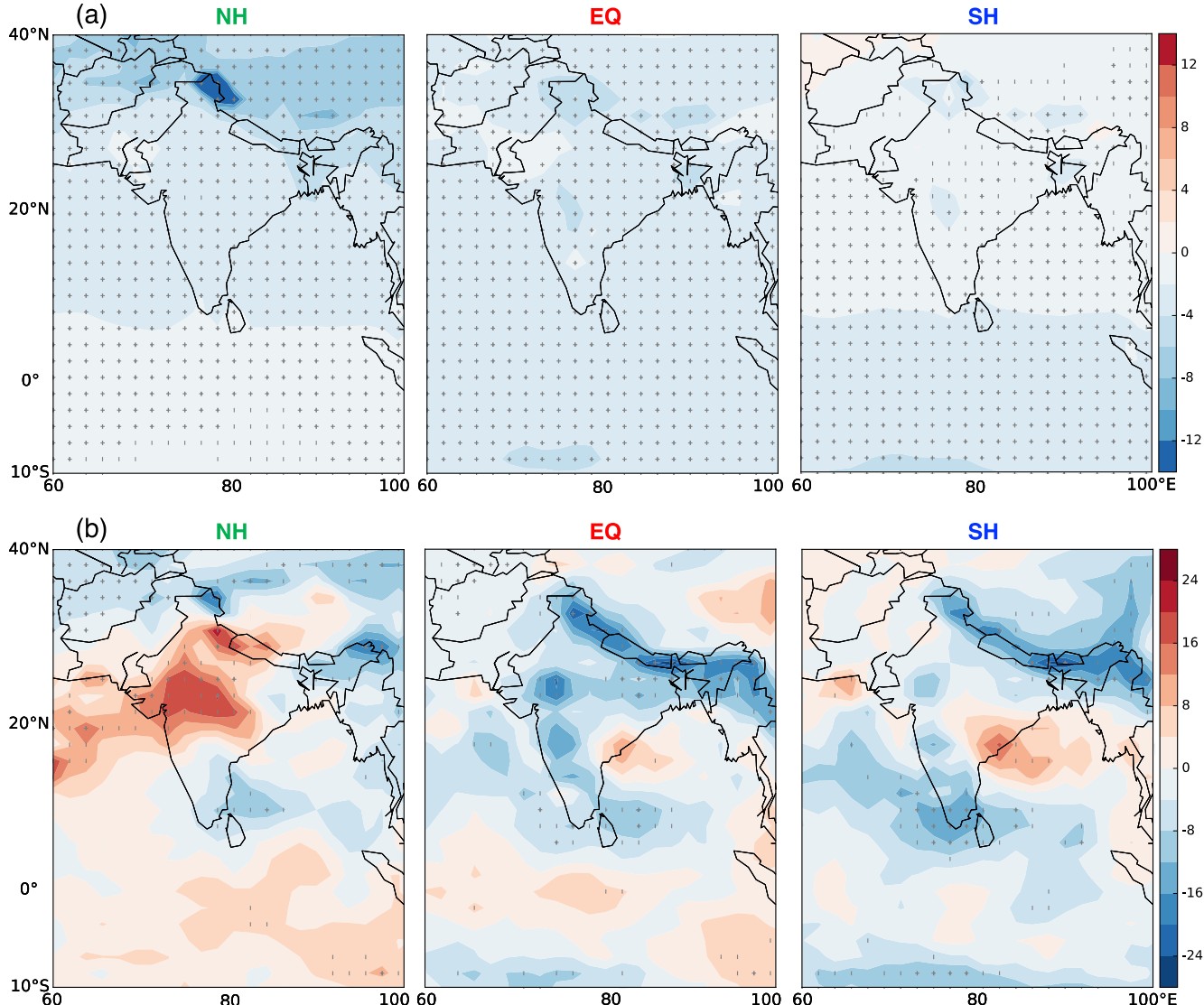

**Figure 10.** Spatial distribution of JJA mean net surface solar radiation (W/m2) anomaly in clear-sky (a) and all-sky conditions (b) after the NH (top), EQ (middle) and SH (bottom) volcanic eruptions in 1992. The vertical bars and plus signs indicate the grid points with significant differences based on the two-tailed student's t-test at the 95% and 99% confidence level, respectively.

decrease of the surface solar radiation is larger in the northern and southern part of the area after the NH and the SH volcanic eruption respectively, while it decreases more uniformly after the EQ eruption. This is due to the reflection of solar radiation by the asymmetric distribution of volcanic aerosols in the two hemispheres after the NH and the SH volcanic eruptions, while being approximately balanced between the two hemispheres after the EQ eruption. However, there are some stronger and opposite regional changes in the all-sky net surface solar radiation (Fig. 10b). Specifically, in India, the net surface solar radiation increases after the NH volcanic eruption, but decreases after the SH and the EQ volcanic eruptions. This indicates

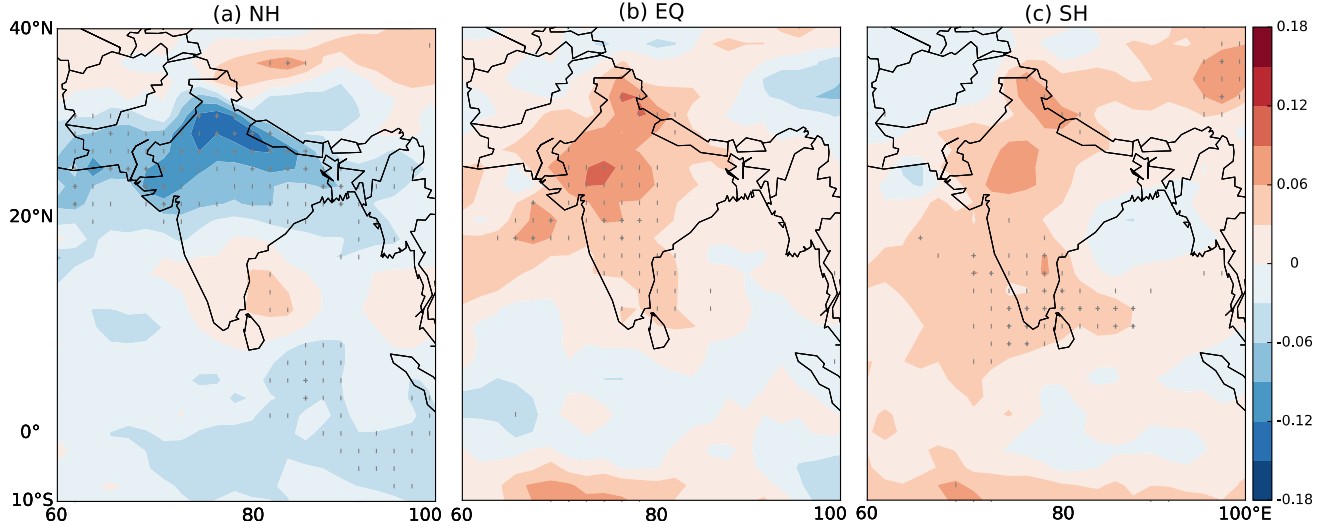

**Figure 11.** Spatial distribution of JJA mean total cloud cover (%) anomaly after the NH (a), EQ (b) and SH (c) volcanic eruptions in 1992. The vertical bars and plus signs indicate the grid points with significant differences based on the two-tailed student's t-tests at the 95% and 99% confidence level, respectively.

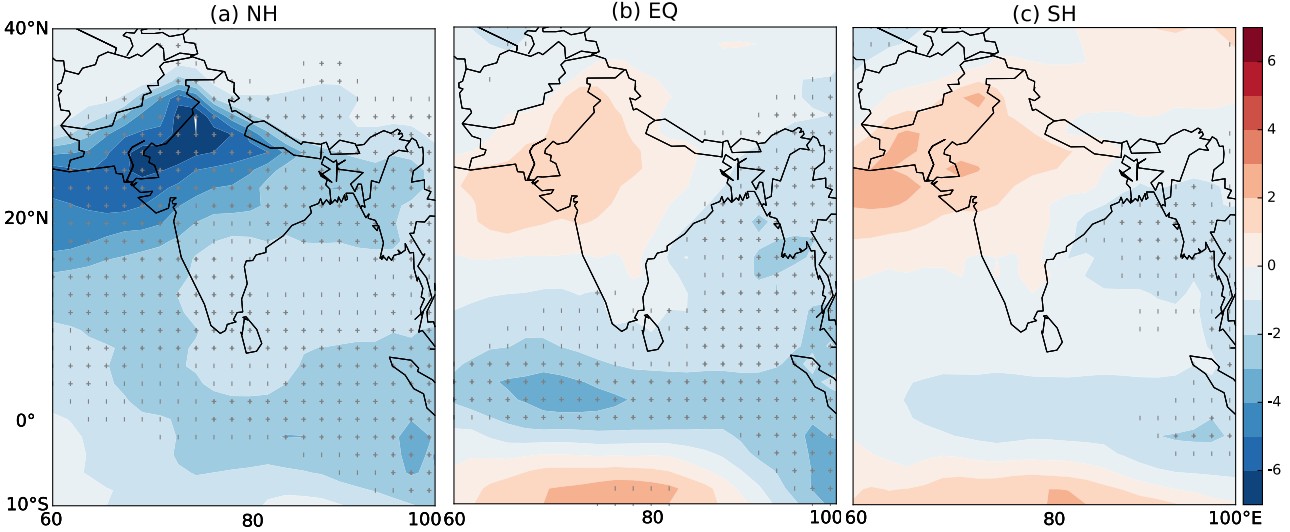

**Figure 12.** Spatial distribution of JJA mean vertical integrated water vapor (kg/m2) anomaly after the NH (a), EQ (b) and SH (c) summer volcanic eruptions in 1992. The vertical bars and plus signs indicate the grid points with significant differences based on two-tailed student's t-tests at the 95% and 99% confidence level, respectively.

that in the regional climate responses to volcanic eruptions, the change of the regional cloud cover plays an important role, leading to a temperature increase in India after the NH volcanic eruption but a temperature decrease after the SH and the EQ eruptions (Fig. 5), which is opposite to the temperature changes in most other northern hemispheric land regions. As shown in Fig. 11, cloud cover decreases significantly after the NH volcanic eruption (Fig. 11a), while it increases after the SH and the EQ volcanic eruptions (Fig. 11c and 11b). Less SR is reflected with less clouds, thus leading to the increase of net

surface solar radiation and temperature after the NH volcanic eruptions. Vice versa, more SR is reflected with more clouds, decreasing net surface solar radiation and temperature after the SH and the EQ eruption.

The dynamical response described in the previous section leads to a reduced moisture transport from the ocean to the land, and the corresponding decrease in the atmospheric water vapor content (Fig. 12) together with the reduced vertical motion reduce cloud formation and precipitation in India after the NH volcanic eruption. However, after the SH and the EQ eruptions, the dynamical response leads to an increased moisture transport and thus increased water vapor content, together with a strengthened vertical motion, enhance cloud formation and precipitation in India.

### 3.5.3 Summary of the response mechanism

Based on the above analyses, the mechanisms behind the different climate effects in India after volcanic eruptions at different latitudes are summarized. The stratospheric aerosols injected by volcanic eruptions directly reflect large amount of solar radiation, which decreases the net surface solar radiation. This leads to cooling in the latitude bands covered by volcanic aerosols. It alters the thermal gradient between the northern and the southern hemisphere and between land and ocean. These changed gradients of available energy and temperature cause subsequent dynamical responses. Specifically, the altered thermal contrast between the two hemispheres moves the ITCZ towards the warmer hemisphere, and the altered land-ocean thermal contrast changes the strength of the South Asian summer monsoon and the associated horizontal and vertical motion of the air. This influences the regional water vapor content and the subsequent formation of clouds. Regional temperature and precipitation are further affected by this change in moisture content and by radiative feedback processes due to the altered cloud cover.

After the NH volcanic eruption, the altered hemispheric thermal contrast leads to a southward movement of ITCZ; the decreased land-ocean thermal contrast weakens the SASM and reduces the horizontal and vertical motion of the air in India. This reduces the atmospheric moisture content and decreases cloud cover, which results in a decrease of the regional precipitation. The reduced cloud cover counteracts the cooling effect of volcanic aerosols. This negative feedback leads to the increase of the regional temperature. Oppositely, after the SH eruption, a northward movement of ITCZ and strengthened SASM strengthen the horizontal and vertical motion. This increases the atmospheric moisture content and cloud cover, thus increasing regional precipitation. The increased cloud cover enhances the cooling of volcanic aerosols, which forms a positive feedback that decreases regional temperature. For the EQ eruption, evenly distributed volcanic aerosols in the two hemispheres cause a relatively symmetric radiative effect in India. However, because the subsequent dynamical response is similar to that after the SH volcanic eruption, the temperature and precipitation patterns are also similar to those after the SH eruption. This illustrates the important roles that dynamical responses and subsequent physical feedbacks play for the regional climate response to volcanic eruptions.

**360  3.6 Discussion**

Our model results indicate significant cooling and precipitation variations especially in the tropical and monsoon regions after volcanic eruptions, in agreement with previous studies (Iles and Hegerl, 2014; Iles et al., 2013). With CESM model simulations, Stevenson et al. (2017) showed different soil moisture responses to volcanic eruptions in April and July. However, temperature and precipitation responses in this study do not show large difference between the summer and winter

eruptions. A different model is used, and different eruption seasons as well as different volcanic forcing with much smaller eruption magnitudes are simulated in our study compared to Stevenson et al. (2017). Toohey et al. (2011), using the MAECHAM5-HAM model, simulated volcanic eruptions with 17 Tg and 700 Tg of $SO_2$ injection in different seasons, and found that the change of all-sky SR is sensitive to eruption season for the 700 Tg of SO2 injection, but is insensitive to eruption season for the Pinatubo-magnitude eruption. The latter is also used in our study. Hence, the sensitivity of the

climate response to volcanic eruption season may be related to the magnitude of the volcanic eruption. Future simulations with different magnitudes and different models will contribute to better understanding this question.

Tropical eruptions were widely assumed to have stronger climate impact than extratropical eruptions (Schneider et al., 2009), until Toohey et al. (2019) pointed out that this might not be the case. Based on the atmosphere-chemistry model MAECHAM5-HAM with fixed sea surface temperature, Toohey et al. (2019) showed a stronger hemispheric cooling after

extratropical explosive volcanic eruptions than after tropical eruptions. Results in this study, based on a fully coupled atmosphere-ocean model, show stronger cooling in the relative hemisphere after the NH and SH eruptions than for the EQ eruptions, supporting the conclusion of Toohey et al. (2019).

Results in this study agree with previous investigations regarding the different climate impacts that interhemispherically asymmetric volcanic aerosol distributions may have in the tropics (Colose et al., 2016; Zuo et al., 2018), the monsoon

regions (Liu et al., 2016; Zhuo et al., 2014; Zuo et al., 2019) and the Atlantic (Yang et al., 2019). These differences are likely caused by the movement of the ITCZ towards the warmer hemisphere with less volcanic aerosol loading (Colose et al., 2016; Haywood et al., 2013; Iles and Hegerl, 2014) and the weakening of monsoonal circulations (Liu et al., 2016; Zuo et al., 2018). Our study confirms these general conclusions in a quantitative way and provides detailed insights into the underlying mechanisms based on model experiments that have been specifically designed to distinguish between NH and SH eruptions.

For the mechanism of the monsoon response to asymmetric volcanic aerosols, Zuo et al. (2019) suggested that the change of the atmospheric circulation plays a dominant role in the change of precipitation, which is related to the changes of the monsoon circulation and the cross-equator flow. Earlier, using the GISS model, Oman et al. (2005) simulated the climate effects of high-latitude eruptions, and found that the radiative effect is larger than the dynamical effect, and the dynamical effect mainly affects the Asian summer, as the strong cooling in northern hemisphere landmass leads to the reduction of the

Asian summer monsoon circulation. This study confirms that the atmospheric circulation change after volcanic eruptions plays an important role in precipitation changes in India, because it changes the water vapor transport and the resultant formation or depletion of clouds in different areas. In addition, regional temperature and precipitation variations are also

affected by radiative cloud feedbacks. This is in agreement with Dogar and Sato (2019), who suggested that the direct radiative effect of tropical volcanic eruption and the associated land-sea thermal contrast result in warming and drying in the

395 Middle Eastern, African and south Asian monsoon regions, which is related to the reduction of clouds over the monsoon region. However, the indirect circulation change was summarized to be connected with the volcanic-induced ENSO forcing in Dogar and Sato (2019). This conclusion might be affected by large uncertainties, as only three ensemble members were used in their study, and the simulations were conducted with an atmospheric circulation model with prescribed oceanic boundary conditions. Our results, based on a fully coupled atmosphere-ocean model and specifically designed to be not

affected by ENSO anomalies at the time of the eruption, show that both the NH and the EQ volcanic eruptions favor an El Niño-tendency in the summer of 1992, while a La Niña-tendency is found after the SH volcanic eruption (Fig. 13). However, after EQ volcanic eruptions, the climate response in India is similar to that after SH eruptions (Fig. 5 and Fig. 6). This suggests that it is the direct circulation change and the subsequent physical feedback, not so much the influence of ENSO, that dominates the regional climate impact in India.

In this study, we simulate volcanic eruptions at different latitudes and in different seasons with the EVA module and MPI-ESM. Limitations exist due to this model setup. EVA is a simplified module that neglects vertical variations of stratospheric dynamics and the impact of the polar vortex, which affects aerosol formation, distribution and aerosol loss especially in a seasonal manner (Toohey et al., 2016). This results in the totally identical $AOD_{550}$ among all the cases (figure 2), which are used as idealized volcanic forcing for our MPI-ESM experiments. This is different from other model simulations with an

interactive aerosol module that can take aerosol microphysical processes and their interaction with cloud processes into account. Aerosols in the stratosphere have a radiative heating effect which can modify the stratospheric dynamics (Marshall et al., 2020). This is not considered in this study as the LR version of MPI-ESM has a limited vertical resolution in the stratosphere and cannot resolve stratospheric dynamics well. These may be the potential reason that the climate response is not sensitive to the eruption season in our study, as the aerosol distribution in the stratosphere depends on the injection

season (Aquila et al., 2012; Tilmes et al., 2017; Visioni et al., 2019). We also do not consider the influence of injection height on the forcing and related climate impact as EVA does not take injection height into consideration. EVA and MPI-ESM model simulations of volcanic eruptions were tuned and validated largely based on observations of the Pinatubo eruption in June 1991. Simulated winter eruptions may thus not be comparably realistic as summer eruptions. However, as pointed out, simulations in this study do not correspond to real volcanic eruptions, and the climate responses cannot be

compared with reality. The focus of this study is the intercomparison of different eruption cases, for which the eruption season and eruption latitude are the only control variables.

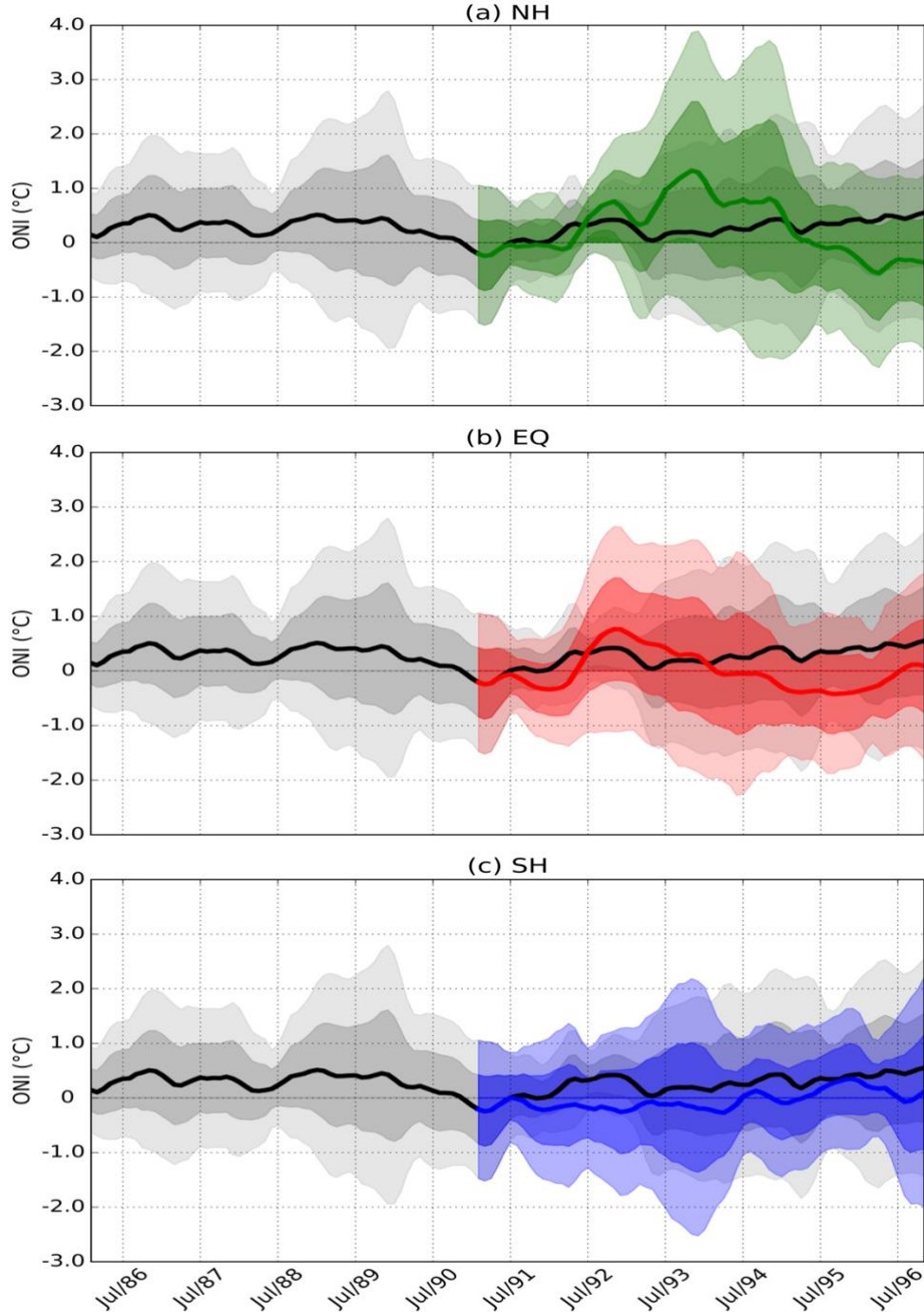

**Figure 13.** Time series of Oceanic Niño index (ºC) anomaly for the NH (a), EQ (b) and SH (c) volcanic eruption cases. The black and relative solid color lines are the ensemble member means of the control run and the eruption cases. The dark (light) grey and relative heavy and light color shades represent one and two standard deviation of the control runs and the eruption cases.

## 4 Summary and conclusions

In order to explore how eruption latitude and eruption season affect the climate impact of volcanic eruptions, model simulations with two groups of Pinatubo-like summer and winter volcanic eruptions at 0° (EQ), 30°N (NH) and 30°S (SH) are performed based on the EVA volcanic forcing generator and the fully coupled general circulation model MPI-ESM. For each experiment case, an ensemble of 10 simulations is performed. Based on the experimental design, results in this study avoid the uncertainties from unequal aerosol magnitudes among different volcanic events and the impact of concurrent ENSO events.

Stratospheric volcanic aerosols reflect incoming solar radiation and reduce the net surface solar radiation. This leads to significant surface cooling, especially in the areas covered by volcanic aerosols. Comparing to equatorial eruptions, volcanic aerosols injected by extratropical eruptions are more concentrated in the specific hemisphere. Because of this, extratropical eruptions in the northern hemisphere (NH cases) cause larger cooling over the northern hemisphere continents, while SH cases cause larger cooling over the southern hemisphere continents, compared to the equatorial eruptions (EQ cases). The precipitation response varies regionally, with stronger responses in the tropic than in the extratropic. Summer eruptions and winter eruptions lead to similar precipitation response patterns, especially in the tropics and monsoon regions, which indicates that the eruption season plays a minor role for the hydrological responses to volcanic eruptions. The largest precipitation change over land occurs in India, one of the most typical monsoon regions. Precipitation is reduced in the Indian monsoon region after the NH volcanic eruptions, but increases after the SH eruptions. The response patterns after the EQ eruptions are similar to those after the SH eruptions, which is likely related to the similar dynamical responses to EQ and SH volcanic eruptions.

Stratospheric volcanic aerosols directly influence the radiative budget of the climate system. This causes dynamical responses due to changes in the available energy and thermal gradients between the northern and the southern hemisphere as well as between land and ocean, associated with interhemispherically asymmetric distributions of volcanic aerosols as well as the different heat capacity of land and ocean.

In India, the NH volcanic eruption leads to warming and drying. This is caused by the decreased water vapor advection and cloud cover over India, which is related to reduced monsoon circulation and convection after the volcanic eruption. The ITCZ moves southward due to the altered interhemispheric thermal contrast. The South Asian summer monsoon weakens due to the decreased land-ocean thermal contrast. After the SH and EQ volcanic eruptions, cooling and wetting result from the increase of water vapor and clouds over India, connected to strengthened horizontal and vertical motions. This is associated with the northward movement of ITCZ and the strengthened South Asian summer monsoon after the volcanic eruptions.

Compared to previous studies, which are usually based on different classifications of historical volcanic eruptions with different numbers of volcanic events, aerosol magnitudes, eruption latitudes and eruption seasons, this study largely reduces these uncertainties by using an idealized model setup. In this setup, the eruption latitude is the only difference between the

three eruption cases, and the eruption season is the only difference between the winter and the summer eruptions. In this way, and by performing 10-member ensemble simulations, we provide robust model results on the sensitivity of the climate effect of volcanic eruptions to eruption latitude and eruption season. These results may also support the interpretation of climate impacts of different choices of stratospheric aerosol injection in geoengineering approaches (Simpson et al., 2019). In contrast to the previous assumption that tropical eruptions have stronger climate effects than extratropical eruptions, this study suggests the opposite. Further research needs to be performed to understand whether tropical or extratropical eruptions have stronger climate impact using different models and experimental setups. In particular, with interactive aerosol modules coupled to higher-resolution models, more processes, like longwave radiation heating due to aerosols and their interaction with cloud cover and stratospheric dynamics, can be investigated. Future simulations can also be performed to investigate whether the sensitivity of the climate response to eruption season depends on the magnitude of volcanic eruptions. The simulations in this study avoid the concurrent effect of volcanic eruptions and ENSO through controlling the initial state of ENSO. Results do not show any significant impact of the eruptions on ENSO variations. The impact of volcanic eruptions on ENSO and their concurrent or different impacts on climate are still unsolved questions, which deserve to be further studied.

**Code availability.** Post-processing and visualization of data was performed with CDO and batch scripts. The scripts are available on request from the corresponding author.

**Data availability.** All data required to reproduce our key results is published by Zhuo et al. (2021) at World Data Center for Climate (WDCC) at Deutsche Klimarechenzentrum (DKRZ).

**Author Contributions.** Z.Z. designed the study, ran the MPI-ESM experiments, analyzed the results and wrote the manuscript. I.K. and U.C. supervised and provided support for designing the study. I.K. provided support for running the experiments. I.K. and S.P. provided support for the analysis. All authors contributed to revising the manuscript.

**Competing interests.** The authors declare that they have no conflict of interest.

**Acknowledgements.** This work is supported by China Scholarship Council (CSC). We thank Matthew Toohey for his help on the EVA module. The authors acknowledge the Deutsche Klimarechenzentrum (DKRZ, https://www.dkrz.de/) for the computational resources and computing facilities. We acknowledge support by the Open Access Publication Fund of the Freie Universität Berlin.

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
