# Peer review of "Climate impact of volcanic eruptions: the sensitivity to eruption season and latitude in MPI-ESM ensemble experiments"

_Atmospheric Chemistry and Physics, 2021_

## Author Response (AR1)

**Response to Reviewers of manuscript**
**"Climate impact of volcanic eruptions: the sensitivity to eruption season and latitude in MPI-ESM ensemble experiments" by Zhihong Zhuo et al.**

We are very grateful to Daniele Visioni and the anonymous reviewer for your kind efforts and thoughtful comments, which are very helpful for enhancing the clarity and quality of the manuscript. We have revised the manuscript carefully according to the reviewers' comments. The list of reviewers' questions and comments (*in italic*) as well as our responses are listed below. Text added to the manuscript is in red. Text removed is in red and struck through.

**1. Response to Reviewer #1**

*Daniele Visioni (Referee)*
*In this work the author analyze the climatic response to tropical and extratropical idealized volcanic eruption, simulating injections at three latitudes and two seasons, and describing the changes at the surface, focusing in particular on surface temperature, precipitation (with a focus on ITCZ shifts and Indian monsoon and cloud cover. The paper is scientifically sound, even if it has a tendency in places to repeat the same, somewhat obvious concept multiple times (like the fact that higher cooling is produced where the aerosols are localized), but it is overall interesting, and the size of the ensembles used allow for some interesting, robust observation. So I think the work can be published on ACP, but the text (and the quality of some figures) needs to be improved in places to make reading it easier. I have offered some suggestions below.*

Thanks for the positive feedback and comments regarding the study. We have revised the text according to the comments and checked carefully before the final submission. The quality of figures will be ensured. They are in PDF format and will be seperately uploaded for the final submission.

***Abstract** Using parenthesis that way makes the abstract unreadable. Better to use two phrases. Please see Robock, 2010 as to why https://eos.org/opinions/parentheses-are-are-not-for-references-and-clarification-saving-space. The same is valid for the whole text in multiple places!*

Thanks for the comment and the reference. We have removed all the parentheses.

*L 18: "emphasizes"*

Revised.

**Introduction**
*L 39: "have been thought" -> "are considered to be"*
*L 60: aims "to answer", no capital letter for "How"*

Revised.

**Method**
*(add an s to "Methods")*

Revised.

*Add information on chemistry used: I assume it's prescribed, but better to specify.*

We added this sentence:
Chemical processes are not explicitly simulated, and background tropospheric and stratospheric aerosols, as part of the forcing data, are represented by their aerosol optical properties (Giorgetta et al., 2013).

*Line 93: which longitude? It's not specified here.*

In this study, the volcanic forcing is generated with the EVA module and then the forcing file is read in when running model simulation with MPI-ESM. This is different from other model simulations with an interactive aerosol module. EVA is a simplified module with emulated time evolution of sulfate mass base in a chemical box model framework in the troposphere and three-boxes in the stratosphere. Each eruption is treated as an instantaneous injection of SO2 into one of the three boxes, with the injection region based on the latitude of the volcano (Toohey et al., 2016). Thus, we can only specify the eruption latitude in the EVA module.

*Line 102: "remains an unsolved question"*

Revised.

*Line 113: maybe add "IVT = " to the equation?*

Added as suggested.

**Results**

*Line 127: see my comment in the Abstract about that use of parenthesis*

We have removed the parentheses that were misused for shortening the sentences.

*Line 130: if the scale in Fig. 2 was a bit narrower (no point in plotting AOD after 1994), one could maybe tell how many months was the "several" said here? Or just specify the e-folding time of the plume.*

Here, "several" is "six", as it reached the maximum in December 1991 (June 1991) for the summer (winter) eruptions" as shown in line 130 in the preprint manuscript. We revised the sentence to:

and is reached after  six months, i.e. in December 1991 and June 1991 for the summer and winter eruptions, respectively.

*Line 135: it depends on the specific of the stratospheric circulation, yes, which are dependent on the season. For instance, see Tilmes et al. (2017) and Visioni et al. (2019) where we tested injections of SO2 in all season at various latitudes in a systematic way with CESM1(WACCM) and the different transport paths are evident.*

We revised the general description of "atmospheric circulation in the tropical stratosphere" to more specified description:

This indicates that the transport of volcanic aerosols from equatorial eruptions to high latitudes depends on the eruption season, which is related to the  large-scale transport of the Brewer Dobson circulation (Hamill et al., 1997).

We also added discussion on the seasonal distribution of stratospheric aerosols in the discussion part, as shown in the following sentences:

These may be the potential reason that the climate response is not sensitive to the eruption season in our study, as the aerosol distribution in the stratosphere depends on the injection season (Aquila et al., 2012; Tilmes et al., 2017; Visioni et al., 2019).

*Line 154: I'm not sure what this confirms, other than the obvious observation that there is less solar radiation at high latitudes during winter?*

Since "confirm" may be misunderstanding, we changed it to "reflect":

This  reflects the role of the seasonal change of the incoming solar radiation in the two hemispheres.

*Line 156: SR is not reflected at the TOA. We see it at the TOA (the model diagnoses it as such) but the SR is reflected in the stratosphere by the volcanic aerosols, thus decreasing SR reaching the surface and cooling the surface.*

We revised the sentence to:

SR  is reflected by the volcanic aerosols in the stratosphere.

*Line 204: aims to explain*

Revised.

*Line 214-217: this entire phrase makes very little sense. Please reprhase. In which case does "the temperature difference between the hemispheres increases slightly"?*

We meant the SH case. As suggested, to make it clearer, we rephrased the sentence to:

Similar results are found in the winter NH and EQ eruption cases,  while the temperature difference between the hemispheres increases slightly in the SH case, indicating a larger cooling in the southern hemisphere than in the northern hemisphere after the winter SH volcanic eruption (Fig. 7b).

*Line 220: how is the ITCZ calculated? There are various ways to do so. Is it the precipitation centroid? Global, or between certain latitudes? Or is it calculated with the position of the Hadley Cell? Please explain.*

In our study, the ITCZ is indicated by the JJA mean zonal mean precipitation between 20° N and 20° S. This was only described in figure 7. To make it clearer, we added the description at the beginning of the Analysis methods section:

The position of the ITCZ is indicated by the latitude of the maximum zonal mean precipitation between 20ºN and 20ºS.

*Section 3.5.2 and 3.5.3 are completely unreadable due to all those parenthesis.*

Sorry for this. We have removed all those parentheses and rephrased the sentences.

*Conclusions*
*Line 355: "Because of this, extratropical eruptions in the northern (the southern) hemisphere (NH and SH cases) cause larger cooling over the northern (southern) hemisphere continents compared to the equatorial eruption (EQ case)" I Have really no idea how to read this phrase: which one is the parenthesis that is supposed to be read first?*

We have revised the sentences to:

Because of this, extratropical eruptions in the northern  hemisphere (NH  cases) cause larger cooling over the northern  hemisphere continents, while SH cases cause larger cooling over the southern hemisphere continents, compared to the equatorial eruptions (EQ cases).

**Figures**
*Some figures (2-3-4) are incredibly low resolution, and they're very hard to understand because of it. Make sure they are higher quality before resubmitting.*

The problem occurred after we merged the pdf figures into the word file. Sorry for the inconvenience, we should have noticed and dealt with the problem. The figures should be clear when separately submitted in pdf format for the final submission.

*Figure 9 would benefit from being more specific in the titles for panels b-d that it is a difference with CTR, and not the full field, even if it says in the caption.*

We pointed out that these are anomalies in Line 118 in the preprint manuscript. And as the referee said, it's clear from the caption. The different scale of color bar should also give a hint. Because all the other figures with EQ, NH, SH in the title also show anomalies, we think it is better to keep it simple and consistent throughout the manuscript.

***References***
*S., Tilmes, H., Richter J., J., Mills M., B., Kravitz, G., MacMartin D., F., Vitt,... F., Lamarque J.â (2017). Sensitivity of aerosol distribution and climate response to stratospheric SO$_2$ injection locations. Journal of Geophysical Research:*
*Atmospheres, 122, 12,591– 12,615. https://doi.org/10.1002/2017JD026888*
*Visioni, D., MacMartin, D. G., Kravitz, B., Tilmes, S., Mills, M. J., Richter, J. H., & Boudreau, M. P. (2019). Seasonal injection strategies for stratospheric aerosol geoengineering. Geophysical Research*
*Letters, 46, 7790– 7799. https://doi.org/10.1029/2019GL083680*

Thanks for the references.

**2. Response to Reviewer #2**

*Anonymous Referee #2*

***General comments***

*This study investigates the impact of volcanic eruptions of Pinatubo magnitude at different latitudes (equator, NH, SH) and in two seasons (winter and summer) on the climate, predominantly on precipitation and the South Asian summer monsoon. The study overall is interesting, and its idealized design allows the mechanisms involved to be investigated systematically, which adds value. The overall novelty could be better signposted and some statements are overstated (see specific comments). More detail on why the summer and winter eruptions have similar impacts despite different forcing would be useful. In general the paper is well written, and the analysis is logical, but the use of parentheses throughout makes some sections difficult to read. I recommend that these are removed. Please also check the quality of figures – e.g. Figure 2 is pixilated and squashed. Overall I think the manuscript will be suited to publication in ACP once these comments have been addressed.*

Thanks for the positive feedback and insightful comments. For the novelty and overstated problem, we have dealt with them, please see the responses to the specific comments. We have removed those parentheses to make it more readable. We improved the figures as suggested, and they will be in pdf format and uploaded separately for the final submission. For the similar impacts of summer and winter eruptions, we added some discussion on the potential reasons. Detailed responses are provided below the related specific comment.

***Specific comments***

*The title is very broad, but the focus is on the South Asian Monsoon, so it would be useful to clarify this. The abstract also refers to 'climate impacts' but it is unclear whether this is in reference just to precipitation. The abstract would also benefit from an overall sentence that highlights the implications of the results.*

For the results and discussion part, section 3.1 to 3.4 showed the global climate response to volcanic eruptions, only section 3.5 focused on South Asian Monsoon. This mechanism exploration part is also used as an example to emphasize that the mechanism is related with both the dynamical and physical feedback. We thus think that the whole content is not properly reflected if we narrow down the title to just focus on the South Asian Monsoon.

In order to emphasize other climate impacts, in the revised manuscript, we pointed out the drying and wetting effect along with the cooling and warming in India. We also added an overall sentence to highlight the implication of the results. Thus, the abstract has been revised to:

**Abstract.** Explosive volcanic eruptions influence near-surface temperature and precipitation especially in the monsoon regions, but the impact varies with different eruption seasons and latitudes. To study this variability, two groups of ensemble simulations are performed with volcanic eruptions in June and December at 0° representing an equatorial eruption (EQ) and at 30° N and 30° S representing northern and southern hemisphere eruptions (NH and SH). Results show significant cooling especially in areas with enhanced volcanic aerosol content. Compared to the EQ eruption, sStronger cooling emerges in the northern (southern) hemisphere after the NH eruption and in the southern hemisphere after the NH (SH) eruption compared to the EQ eruption. Stronger precipitation variations occur in the tropics than in the high latitudes. Summer and winter eruptions lead to similar hydrologicalclimate impacts. The NH and the SH eruptions have reversed climate impacts, especially in the regions of the South Asian summer monsoon (SASM) regions. After the NH (SH) eruption, direct radiative effects of volcanic aerosols induce changes in the interhemispheric and land-sea thermal contrasts, which move the intertropical convergence zone (ITCZ) southward (northward) and weaken (strengthen) the South Asian summer monsoonSASM. This reduces (increases) the moisture transport from the ocean to India, and reduces (enhances) cloud formation and precipitation in India. The subsequent radiative feedbacks due to regional cloud cover lead to warming (cooling) in India. After the SH eruption, vice versa, a northward movement of the ITCZ and strengthening of the SASM, along with enhanced cloud formation, lead to enhanced precipitation and cooling in India. This emphasizes the sensitivity of regional climate impacts of volcanic eruptions to eruption latitude, which relates to the dynamical response of the climate system to radiative effects of volcanic aerosols and the subsequent regional physical feedbacks. Our results indicate the importance of considering dynamical and physical feedbacks to understand the mechanism behind regional climate responses to volcanic eruptions, and may also shed light on the climate impact and potential mechanisms of stratospheric aerosol engineering.

*L28 - It is well known that the spatial distribution of volcanic aerosols affects the climate impact, and these studies are not the first to show this.*
    We added two references and revised the sentences to:
However,  the spatial distribution of volcanic aerosols and associated radiative forcing affects the climate impact of volcanic eruptions (Robock, 2000; Timmreck, 2012; Toohey et al., 2019; Yang et al., 2019).

*L33 – There are many other studies that have highlighted the importance of season on the climate impact (e.g. Toohey et al. 2011; 2013; 2016a; Aquila et al., 2012; Stoffel et al., 2015). It would be useful here to state what Stevenson et al. (2017) found in relation to the role of season. What about studies that have investigated the importance of latitude?*
    We added several suggested references and also refered to the impact of eruption latitude. We didn't describe in detail what Stevenson et al. (2017) found here as it is discussed in the discussion part in line 306-310 in the preprint manuscript. We think it's better to keep this part on a more general level and revised the sentence to:
Furthermore,  the climate impact of volcanic eruptions is affected by eruption latitude (Marshall et al., 2020; Yang et al., 2019; Zuo et al., 2019) and eruption season (Aquila et al., 2012; Stevenson et al., 2016; Toohey et al. 2011; 2013).

*L39 – What is the difference between extratropical and high-latitude here?*

Icelandic eruptions were considered when referring to high latitude eruptions, but as pointed out here, actually, "extratropical" includes the high latitudes, so we removed high-latidude and revised the sentence to:

Tropical eruptions are considered to have larger climate impacts than extratropical  eruptions (Myhre et al., 2013; Schneider et al., 2009).

*L40 – Satellite observations for tropical eruptions show this spread so I do not think 'believe' is the right word here.*

"believe" was revised to "can", we also added two references. We revised the sentence to:

The volcanic aerosols injected into the stratosphere from a tropical eruption can be transported to both hemispheres and finally reach both poles (Robock, 2000; Aquila et al., 2012).

*L47 - 49 – I am not sure controversial is the right word. It also depends on what 'climate impact' is being investigated and this is specific to NH cooling.*

As written in the original manuscript, Schneider et al. (2009), Myhre et al. (2013) and Toohey et al. 2019 had different conclusions. We revised the sentence to be:

Thus, previous studies came to different conclusions on whether tropical or extratropical volcanic eruptions have larger climate impact.

*L65 - It would be useful to introduce to the reader that you first explore the global forcing and climate response (sections 3.1 to 3.4), before focusing on the precipitation response in India (section 3.5)*

As suggested, we added the following sentence:

We first show the global forcing and climate responses to volcanic eruptions in section 3.1 to 3.4, and then focus on the mechanism of the precipitation response in India in section 3.5.

*L73 – Are there any implications of using a low-resolution version for looking at regional climate? Is this important?*

Older versions of this model were used for studies of the Asian monsoon and its response to volcanic eruptions by Guo et al., 2016; Man et al., 2012; 2014; Zhang et al, 2012 etc. This low-resolution (LR) version has been improved compared to its older version and is widely used, e.g., for CMIP6. In addition, although we look at regional climate, we still focus on large scale circulation patterns, i.e. ITCZ and SASM when exploring the mechanism of climate response in India. For the ensemble approach adopted in this study, it would be very expensive to apply a higher-resolution model version, and it is beyond the scope of this study to investigate if such a higher-resolution model would produce more realistic results.

*L92 – It would be useful here to state that these are meteorological ensemble members or to introduce the 10 members only at the end of this section.*

Considering the suggestion, we moved the sentences to the end of this paragraph.

*L98 – Why 23 control runs? Is there a reason behind this? Figure 1 is somewhat confusing with the branches at the bottom for the 23 control runs – is this necessary?*

Yes, after 23 control runs, we finally picked out 10 ensembles members with neutral ENSO initial conditions. As they obviously led to confusion, we removed the bottom branches of figure 1.

*L104 – Six sets of ensembles not three?*

We were referring to three sets for each winter and summer eruptions, but it's confusing with parenthesis, so we revised the sentence to:

Restart files from these 10 control runs are used to initialize  six 10-member ensemble simulations  of EQ, NH and SH eruptions in summer and winter for the period of 1991-1996 and 1990-1996, respectively.

*L122 – This line seems to undermine the purpose of the study. If it is not an equal focus, then I recommend it be removed from the title or that both seasons are investigated equally, and the limitations added to the discussion.*

"More focus on summer cases" is related to the final part where we only use the summer eruptions cases to explain the response mechanism in India. We pointed out this with the reason in line 241-243 in the preprint manuscript. As suggested, we removed this sentence and added the following sentences to discuss the limitations:

EVA and MPI-ESM model simulations of volcanic eruptions were tuned and validated largely based on observations of the Pinatubo eruption in June 1991. Simulated winter eruptions may thus not be comparably realistic as summer eruptions. However, as pointed out, simulations in this study do not correspond to real volcanic eruptions, and the climate responses cannot be compared with reality. The focus of this study is the intercomparison of different eruption cases, for which the eruption season and eruption latitude are the only control variables.

*L128 – Is it realistic that the global mean time series are the same? It would be useful to add to the discussion some of the limitations of EVA.*

Good point, it's not realistic as it ignores the aerosol microphysics and their related feedbacks on the forcing. As suggested, we added the following sentences to discuss the limitations of EVA:

EVA is a simplified module that neglects vertical variations of stratospheric dynamics and the impact of the polar vortex, which affects aerosol formation, distribution and aerosol loss especially in a seasonal manner (Toohey et al., 2016). This results in the totally identical $AOD_{550}$ among all the cases (figure 2), which are used as idealized volcanic forcing for our MPI-ESM experiments. This is different from other model simulations with an interactive aerosol module that can take aerosol microphysical process and their interaction with cloud processes into account.

*L135 – consider adding 'reflecting the large-scale transport of the Brewer Dobson circulation' or similar.*

As suggested, we removed "atmospheric circulation in the tropical stratosphere" and revised the sentence to:

This indicates that the transport of volcanic aerosols from equatorial eruptions to high latitudes depends on the eruption season, which is related to the  large-scale transport of the Brewer Dobson circulation (Hamill et al., 1997).

*L154 – Also recently discussed in Marshall et al. (2020) for a wide range of eruption latitudes. What about the LW heating following these eruptions and the impact on stratospheric dynamics and also on cloud cover? It may be useful to look at the surface radiation changes.*

Yes, as discussed in Marshall et al. (2020), the net radiative forcing depends on the volcanic AOD, insolation, cloud cover and surface albedo, which can be explained by the

differences in the large-scale stratospheric circulation and aerosol microphysical processes. The aerosol radiative heating can modify the stratospheric dynamics. However, the model used in this study cannot resolve stratospheric dynamics well, and the EVA forcing used cannot involve aerosol microphysical processes, these limitations make it hard to explore these details. Future simulations with an interactive aerosol module coupled to a higher resolution model will help to investigate these factors. We added some discussions in the discussion and conclusion part with the following sentences:

This is different from other model simulations with an interactive aerosol module that can take aerosol microphysical process and their interaction with cloud processes into account. Aerosols in the stratosphere have a radiative heating effect which can modify the stratospheric dynamics (Marshall et al., 2020). This is not considered in this study as the LR version of MPI-ESM has a limited vertical resolution in the stratosphere and cannot resolve stratospheric dynamics well.

In particular, with interactive aerosol modules coupled to higher-resolution models, more processes, like longwave radiation heating due to aerosols and their interaction with cloud cover and stratospheric dynamics, can be investigated.

*L172 – It would be useful to state the number of months after the eruption when the max cooling occurs in addition to the date.*

Considering this comment, we thought it's better to add the number of months to reach the maximum cooling after the eruptions in the former paragraph where describing figure 4. Thus the sentences there has been revised to:

For the summer eruption cases, it takes 15, 12 and 15 months to reach the maximum cooling after the NH, the EQ and the SH eruption, respectively, thus stronger cooling emerges after the NH eruption than the SH and EQ eruptions, and the coolest boreal summer among all the cases occurs in the boreal summer of 1992 (Fig. 4a). Most significant cooling emerges in the northern hemisphere mid-latitudes in the NH case, in the tropics in the EQ case and in the southern hemisphere mid-latitude areas in the SH case, as indicated by the stippling in Fig. 4c. For the winter eruption cases, a significant and strong cooling is visible in the boreal summer of 1991; the maximum cooling shows 11 months after the NH eruption (Fig. 4b) and emerges in the northern hemisphere mid-latitude areas (Fig. 4d). The maximum cooling occurs 16 months after the EQ eruption in the boreal spring of 1992 (Fig. 4b), and the abnormal cooling is pronounced in the tropics as indicated by the stippling in Fig. 4d. A similar cooling trend with a smaller magnitude is shown 16 months after the SH eruption (Fig. 4b), but the cooling is significant in the southern hemisphere mid-latitudes (Fig. 4d).

*Figure 5 – please make the stippling lighter as it is hard to see the underlying temperature anomalies or stipple the areas that are not significant.*

As suggested, we made the stippling lighter in the revised figure.

*L177 – but these ocean changes aren't significant?*

We can't fully understand this comment. As shown in fig. 5, there is a lot of stippling over the ocean, especially in those areas with strong cooling effects, indicating the significance of the cooling over the ocean.

*L190-201 – can you point out some of these numbers and highlight what the reversed responses are? I do not see the black box in fig 6 and a small scale is present under panel d.*

It's nice to point out some numbers to indicate the magnitude of changes with time series analysis, but as pointed out in the preprint manuscript: "Time series of global mean precipitation from our experiments do not show significant changes (not shown), but there

are precipitation responses on a regional level. Thus, here we only show the spatial figures and think it is better to focus on the changes of spatial pattern instead of the magnitude.

We added a description to explain the reversed precipitation responses:
In many tropical regions, the NH (Fig. 6a and 6b) and SH (Fig. 6e and 6f) volcanic eruptions lead to reversed precipitation responses, i.e. opposite decrease and increase of precipitation up to over 3 mm/day in some areas.

The reversed responses are shown in India and described in the preprint manuscript: The most pronounced changes over land are found in India, which is one of the most typical monsoon regions. Precipitation is reduced in India after the NH eruptions (Fig. 6a and 6b), but strongly increased after the EQ and SH volcanic eruptions (Fig. 6c to 6f).

Sorry we forgot to remove the text about the black box in fig. 6. We thought to mark out India with a black box, but later found that it's not necessary, as India is obvious enough. We have removed this text in the revised manuscript.

The black line under panel d has been removed in fig. 6.

*Figure 7 c and d – Why is the control line different in panels c and d? Please label the y axes and indicate that the control line is not an anomaly and shown on the secondary y axis.*

Because it shows the ITCZ in 1992 for the summer eruptions (c) and in 1991 for the winter eruptions (d). Sorry, the description of the figure caption was not accurate, we revised the years to be connected with panels c and d. The figure caption has been revised as follows:
Figure 7: Difference of JJA mean temperature (ºC) anomaly between the northern and the southern hemisphere after the summer eruptions  (a) and the winter eruptions  (b), and the position of the ITCZ in 1992 (c) and 1991 (d) indicated by the JJA mean zonal mean precipitation (mm/day) anomaly between 20°N and 20°S. The grey bar in (a) and (b) indicates the two-standard deviation of the control runs.

We added labels for both y-axes in figure 7c and 7d as suggested.

*Sections 3.5.2 and 3.5.3 are very difficult to read with the parentheses. Why are the summer and winter responses similar despite different forcing?*

Sorry for the difficulties. We removed all those parentheses and rephrased the text.

The similar responses to summer and winter eruptions can be due to the small eruption magnitude simulated in our study. As Toohey et al. (2011) pointed out that although eruption season has a significant influence on AOD and clear-sky SW radiation anomalies, all-sky SW anomalies are insensitive to eruption season for the Pinatubo-magnitude eruption, but is sensitive for larger eruption magnitudes. The similar responses may also result from the simplicity of EVA, which produced absolutely identical global mean AOD values in all cases (as shown in figure 1). The similar responses may also be connected to limitations of model setup with EVA and MPI-ESM, as it cannot resolve interactive aerosol module and stratospheric dynamics due to limited vertical resolution in the stratosphere, and thus cannot properly represent related processes like aerosol radiative heating and its impact on stratospheric dynamics. We added following discussions on these factors:
A different model is used, and different eruption seasons as well as different volcanic forcing with much smaller eruption magnitudes are simulated in our study compared to Stevenson et al. (2017). Toohey et al. (2011), using the MAECHAM5-HAM model, simulated volcanic eruptions with 17 Tg and 700 Tg of $SO_2$ injection in different seasons, and found that the change of all-sky SR is sensitive to eruption season for the 700 Tg of SO2 injection, but is insensitive to eruption season for the Pinatubo-magnitude eruption. The latter is also used in our study. Hence, the sensitivity of the climate response to volcanic eruption season may be related to the magnitude of the volcanic eruption. Future simulations with different magnitudes and different models will contribute to better understanding this question.

In this study, we simulate volcanic eruptions at different latitudes and in different seasons with the EVA module and MPI-ESM. Limitations exist due to this model setup. EVA is a simplified module that neglects vertical variations of stratospheric dynamics and the impact of the polar vortex, which affects aerosol formation, distribution and aerosol loss especially in a seasonal manner (Toohey et al., 2016). This results in the totally identical AOD550 among all the cases (figure 2), which are used as idealized volcanic forcing for our MPI-ESM experiments. This is different from other model simulations with an interactive aerosol module that can take aerosol microphysical process and their interaction with cloud processes into account. Aerosols in the stratosphere have a radiative heating effect which can modify the stratospheric dynamics (Marshall et al., 2020). This is not considered in this study as the LR version of MPI-ESM has a limited vertical resolution in the stratosphere and cannot resolve stratospheric dynamics well. These may be the potential reason that the climate response is not sensitive to the eruption season in our study, as the aerosol distribution in the stratosphere depends on the injection season (Aquila et al., 2012; Tilmes et al., 2017; Visioni et al., 2019).

*L273 – not just the change in the cloud cover, but also the background cloud compared to clear-sky conditions? What about the role of surface cooling?*
        Figure 10 shows the anomaly of the net surface solar radiation with respect to the control run without any volcanic eruption. This anomaly extracts the potential background cloud without volcanic impact. Thus, it reflects the cloud cover changes when comparing the full-sky anomalies with the clear-sky anomalies.
        For the surface cooling, as discussed in section 3.5.1, the temperature changes in different regions lead to hemispheric and land-sea thermal contract, which changes the circulation patterns and the transport of water vapor. These dynamical effects change the formation of the cloud in different regions.

*L310 – Stevenson et al. (2017) also looked at much larger eruptions with very different forcing.*
        True, we added this to the sentence:
A different model is used, and different eruption seasons as well as different volcanic forcing with much smaller eruption magnitudes are simulated in our study compared to Stevenson et al. (2017).

*L342 – Because the ENSO response is mentioned in the conclusion, this result should at least be shown in the SI.*
        We added the ENSO figure as fig. 13 in the revised manuscript.

*L365 – What about other dynamical effects, such as aerosol heating and changes to the polar vortex and other circulation changes. Is this important?*
        Yes, they are important for understanding the actual comprehensive climate impacts of volcanic eruptions, but it's hard to investigate these factors in this study, because we don't have an interactive aerosol module in our model simulation with MPI-ESM and EVA. We added some discussions on these factors connected to the limitation of this model setup, as shown by following sentences:
EVA is a simplified module that neglects vertical variations of stratospheric dynamics and the impact of the polar vortex, which affects aerosol formation, distribution and aerosol loss especially in a seasonal manner (Toohey et al., 2016). This results in the totally identical

AOD550 among all the cases (figure 2), which are used as idealized volcanic forcing for our MPI-ESM experiments. This is different from other model simulations with an interactive aerosol module that can take aerosol microphysical process and their interaction with cloud processes into account. Aerosols in the stratosphere have a radiative heating effect which can modify the stratospheric dynamics (Marshall et al., 2020). This is not considered in this study as the LR version of MPI-ESM has a limited vertical resolution in the stratosphere and cannot resolve stratospheric dynamics well.

*Conclusions: some statements are fairly obvious, and I do not think necessary e.g. 'results confirm that aerosols reflect incoming solar radiation'. Limitations to this study are also missing from the discussion/conclusions. For example how would your results differ with a larger injection magnitude? Would season and latitude become more important? If EVA is tuned for a summer eruption how robust are these results for winter? The temperature response (e.g. Figure 4) is also different for winter vs. summer eruptions but the conclusion is very general in saying that season plays a minor role in climate impacts. It needs to be clear this is in relation to precipitation change.*

Since it is an obvious conclusion, but is connected with the following sentence, we revised the sentences to:

Stratospheric volcanic aerosols  reflect incoming solar radiation and reduce the net surface solar radiation. This leads to significant surface cooling, especially in the areas covered by volcanic aerosols.

We added some discussions on limitations in the discussion part, and also some related outlook of future work in the conclusion part, as shown by the following sentences:

In this study, we simulate volcanic eruptions at different latitudes and in different seasons with the EVA module and MPI-ESM. Limitations exist due to this model setup. EVA is a simplified module that neglects vertical variations of stratospheric dynamics and the impact of the polar vortex, which affects aerosol formation, distribution and aerosol loss especially in a seasonal manner (Toohey et al., 2016). This results in the totally identical AOD550 among all the cases (figure 2), which are used as idealized volcanic forcing for our MPI-ESM experiments. This is different from other model simulations with an interactive aerosol module that can take aerosol microphysical process and their interaction with cloud processes into account. Aerosols in the stratosphere have a radiative heating effect which can modify the stratospheric dynamics (Marshall et al., 2020). This is not considered in this study as the LR version of MPI-ESM has a limited vertical resolution in the stratosphere and cannot resolve stratospheric dynamics well. These may be the potential reason that the climate response is not sensitive to the eruption season in our study, as the aerosol distribution in the stratosphere depends on the injection season (Aquila et al., 2012; Tilmes et al., 2017; Visioni et al., 2019). We also do not consider the influence of injection height on the forcing and related climate impact as EVA does not take injection height into consideration. EVA and MPI-ESM model simulations of volcanic eruptions were tuned and validated largely based on observations of the Pinatubo eruption in June 1991. Simulated winter eruptions may thus not be comparably realistic as summer eruptions. However, as pointed out, simulations in this study do not correspond to real volcanic eruptions, and the climate responses cannot be compared with reality. The focus of this study is the intercomparison of different eruption cases, for which the eruption season and eruption latitude are the only control variables.

In particular, with interactive aerosol modules coupled to higher-resolution models, more processes, like longwave radiation heating due to aerosols and their interaction with cloud cover and stratospheric dynamics, can be investigated. Future simulations can also be performed to investigate whether the sensitivity of the climate response to eruption season depends on the magnitude of volcanic eruptions.

We meant precipitation response patterns, but the climate impact was misleading. To make it clearer, we revised the sentence to:

Summer eruptions and winter eruptions lead to similar precipitation response patterns, especially in the tropics and monsoon regions, which indicates that the eruption season plays a minor role for the  hydrological responses to volcanic eruptions.

***Technical corrections***
*L3 – earth --> Earth*
*L39 - impacts*
*L47 – simulations*
*L102 – remove 'to be' L241 – since --> because L357 – tropics, extratropics*
Revised.

***References***
*Aquila, V., Oman, L.D., Stolarski, R.S., Colarco, P.R. and Newman, P.A. 2012. Dispersion of the volcanic sulfate cloud from a Mount Pinatubo-like eruption. Journal of Geophysical Research-Atmospheres. 117, D06216*

*Marshall, L. R., Smith, C. J., Forster, P. M., Aubry, T. J., Andrews, T., and Schmidt, A. 2020. Large variations in volcanic aerosol forcing efficiency due to eruption source parameters and rapid adjustments. Geophysical Research Letters, 47, e2020GL090241.*

*Stoffel, M., Khodri, M., Corona, C., Guillet, S., Poulain, V., Bekki, S., Guiot, J., Luckman, B.H., Oppenheimer, C., Lebas, N., Beniston, M. and Masson-Delmotte, V. 2015. Estimates of volcanic-induced cooling in the Northern Hemisphere over the past 1,500 years. Nature Geoscience. 8(10), 784-788.*

*Toohey, M., Kruger, K., Niemeier, U. and Timmreck, C. 2011. The influence of eruption season on the global aerosol evolution and radiative impact of tropical volcanic eruptions. Atmospheric Chemistry and Physics. 11(23), 12351-12367.*

*Toohey, M., Kruger, K., Sigl, M., Stordal, F. and Svensen, H. 2016a. Climatic and societal impacts of a volcanic double event at the dawn of the Middle Ages. Climatic Change. 136(3-4), 401-412.*

*Toohey, M., Kruger, K. and Timmreck, C. 2013. Volcanic sulfate deposition to Greenland and Antarctica: A modeling sensitivity study. Journal of Geophysical Research-Atmospheres. 118(10), 4788-4800.*

Thanks for the references.